# Long non-coding RNAs are essential for *Schistosoma mansoni* pairing-dependent adult worm homeostasis and fertility

**Gilbert O. Silveira**[1,2], **Helena S. Coelho**[1], **Adriana S. A. Pereira**[1,2], **Patrícia A. Miyasato**[3], **Daisy W. Santos**[1,2], **Lucas F. Maciel**[1], **Giovanna G. G. Olberg**[1], **Ana C. Tahira**[1], **Eliana Nakano**[3], **Maria Leonor S. Oliveira**[4], **Murilo S. Amaral**[1]*, **Sergio Verjovski-Almeida**[1,2]*

**1** Laboratório de Ciclo Celular, Instituto Butantan, São Paulo, São Paulo, Brazil, **2** Instituto de Química, Universidade de São Paulo, São Paulo, São Paulo, Brazil, **3** Laboratório de Parasitologia, Instituto Butantan, São Paulo, São Paulo, Brazil, **4** Laboratório de Bacteriologia, Instituto Butantan, São Paulo, São Paulo, Brazil

* murilo.amaral@butantan.gov.br (MSA); sergio.verjovski@butantan.gov.br (SVA)

**Data Availability Statement:** The authors confirm that all data underlying the findings are fully available without restriction. All relevant data are

## Abstract

The trematode parasite *Schistosoma mansoni* causes schistosomiasis, which affects over 200 million people worldwide. Schistosomes are dioecious, with egg laying depending on the females' obligatory pairing with males. Long non-coding RNAs (lncRNAs) are transcripts longer than 200 nucleotides with low or no protein-coding potential that have been involved in other species with reproduction, stem cell maintenance, and drug resistance. In *S. mansoni*, we recently showed that the knockdown of one lncRNA affects the pairing status of these parasites. Here, we re-analyzed public RNA-Seq data from paired and unpaired adult male and female worms and their gonads, obtained from mixed-sex or single-sex cercariae infections, and found thousands of differentially expressed pairing-dependent lncRNAs among the 23 biological samples that were compared. The expression levels of selected lncRNAs were validated by RT-qPCR using an *in vitro* unpairing model. In addition, the *in vitro* silencing of three selected lncRNAs showed that knockdown of these pairing-dependent lncRNAs reduced cell proliferation in adult worms and their gonads, and are essential for female vitellaria maintenance, reproduction, and/or egg development. Remarkably, *in vivo* silencing of each of the three selected lncRNAs significantly reduced worm burden in infected mice by 26 to 35%. Whole mount *in situ* hybridization experiments showed that these pairing-dependent lncRNAs are expressed in reproductive tissues. These results show that lncRNAs are key components intervening in *S. mansoni* adult worm homeostasis, which affects pairing status and survival in the mammalian host, thus presenting great potential as new therapeutic target candidates.

## Author summary

*Schistosoma mansoni* is a parasite that causes schistosomiasis, affecting over 200 million people worldwide. Only one drug is available for treatment, and new therapeutic strategies

within the paper and its Supporting Information files.

**Funding:** This work was supported by a grant from Fundação de Amparo à Pesquisa do Estado de São Paulo (FAPESP) Thematic grant number 2018/23693-5 to S.V.A. G.O.S., D.W.S., L.F.M., and A.S. A.P. received fellowships from FAPESP (18/24015-0, 19/09404-3, 18/19591-2, and 16/10046-6, respectively). S.V.A. received an established investigator fellowship award from Conselho Nacional de Desenvolvimento Científico e Tecnológico (306646/2019-6), Brasil. The funders had no role in study design, data collection and analysis, decision to publish, or preparation of the manuscript.

**Competing interests:** The authors have declared that no competing interests exist.

are required. *S. mansoni* females need to be continuously paired with males to be sexually mature and to produce eggs that cause tissue pathology. Therefore, interfering with adult worm pairing may provide new therapeutic intervention alternatives. However, the entire set of factors driving this pairing status are not fully understood. Long non-coding RNAs (lncRNAs) are transcripts longer than 200 nucleotides with low or no protein coding potential. They have been shown in other species to be involved with reproduction. Here, we show that lncRNAs are essential for *S. mansoni* pairing-dependent adult worm homeostasis and/or fertility. We found that hundreds of lncRNAs are differentially expressed between paired and unpaired adult worms. We show by *in vitro* and *in vivo* gene silencing approaches that lncRNAs are essential to regulate the cell proliferation status of adult worm and their gonads, and to maintain worm pairing, female reproductive capacity, and/or egg development. Our study shows that lncRNAs are key components intervening in *S. mansoni* adult worm homeostasis, which affects pairing status and survival, thus presenting great potential as new therapeutic target candidates.

## Introduction

Schistosomiasis is a neglected tropical disease that affects more than 200 million people worldwide [1–3]. No vaccine has been developed so far and controlling the disease is still a challenge, with treatment using a single drug, praziquantel [4, 5]. Furthermore, praziquantel-resistant strains have been reported and the drug is effective only against the adult stage of the parasite [6, 7]. Thus, understanding schistosome biology on a molecular level is needed and could suggest new therapeutic alternatives [8, 9].

*Schistosoma mansoni* is the species present in the Americas and Africa, with adult worms living in the mesenteric veins of the mammalian host. Schistosomes are the only mammalian trematodes that are dioecious, possessing male and female separate sexes [10]. Once paired, female egg production is initiated and results in the daily production of 300–3,000 eggs per female, depending on the species [11]. Approximately half of the eggs reach the gut lumen (most *Schistosoma* species) or the bladder (*S. haematobium*). The remaining eggs are swept away via the blood system mainly into the liver and spleen, where they penetrate the tissues causing severe inflammation and liver cirrhosis, the main cause of mortality.

Male and female adult worms must stay paired together throughout their life for reproduction to be successful. In fact, female sexual development is governed by male pairing, meaning that females that are not paired to males have immature reproductive status and thus produce no eggs [12–16]. RNA-Seq analyses of adult worms and their gonads retrieved from mammalian hosts infected with mixed-sex or single-sex cercariae have paved the way for understanding the role of protein-coding genes in the maintenance of the pairing status in schistosomes, with the identification of molecular pathways involved in sexual development [14, 17–19]. Recently, a male-derived non-ribosomal dipeptide pheromone that controls female schistosome sexual development and egg laying has been described [20]. However, the complete set of molecular players that intervene in sexual development are not fully characterized.

Long non-coding RNAs (lncRNAs) are RNAs longer than 200 nucleotides with low or no protein-coding potential that have been implicated in regulating different biological processes in humans and in many other species [21]. In mammalians, lncRNAs work in various cellular environments and thus can act as regulators of protein-coding gene expression [22, 23], stem cell maintenance [24], and drug resistance [21]. Due to their tissue-specific expression and multifaceted functions, lncRNAs were proposed as novel therapeutic targets in human diseases

[25, 26]. In addition, they have been suggested as potential targets of antiparasitic therapies, as reviewed by Silveira et al., 2022 [27]. Recently, single-cell RNA-seq analyses showed that lncRNAs are conspicuously expressed in *S. mansoni* gamete and tegument progenitor cell populations [28]. In fact, lncRNAs have been associated with epigenetic drug treatment in *S. mansoni* [29], and knockdown of one selected lncRNA has been shown to interfere with *in vitro* parasite pairing, female vitellaria development and adult worm stem cell proliferation [27]. Nevertheless, very little is known regarding lncRNAs and their relationship with pairing status and/or fertility of adult worms.

Because lncRNAs have been proposed as possible regulators of many biological processes we hypothesized that lncRNAs could be differentially expressed between sexually immature and mature worms. Here, we report that lncRNAs are differentially expressed between paired and unpaired adult *S. mansoni* worms. We show by *in vitro* and *in vivo* silencing experiments that lncRNAs are essential for maintaining the pairing status, worm viability, female reproductive performance, and/or adult worm stem cell proliferation as well as for parasite survival in infected mice. Using whole mount *in situ* hybridization (WISH), we show that selected lncRNAs are expressed in reproductive organs. We propose lncRNAs as new potential therapeutic target candidates against schistosomiasis since they are key components of adult worm pairing status in *S. mansoni*.

## Results

### Identification of lncRNAs differentially expressed between immature and mature *S. mansoni* parasites and their gonads

In order to identify the complement of long non-coding RNAs (lncRNAs) expressed in sexually immature and mature *S. mansoni* parasites, we re-analyzed RNA-Seq data generated by Lu et al., 2016 [14] who studied parasites retrieved from hamsters infected with either mixed-sex cercariae (b, bisex or mixed-sex infections) or cercariae of only a single sex (s, single-sex infections) (**Fig 1A**). Lu et al. compared eight different biological conditions, namely paired whole males (M) and their testes (T) from mixed-sex (bM and bT) infections, unpaired males and their testes from single-sex (sM and sT) infections, paired whole females (F) and their ovaries (O) from mixed-sex (bF and bO) infections, or unpaired females and their ovaries from single-sex (sF and sO) infections (**Fig 1A**). Note that we kept the prefix "b" for the mixed-sex infections, to be consistent with the nomenclature of the original work [14], in which "b" was used for bisex infection.

The original RNA-Seq analysis of Lu et al., 2016 [14] only considered the protein-coding genes that were expressed among all processed samples [14]. Our re-analyses of those data identified the expression of 16,583 lncRNAs in mixed-sex/single-sex *S. mansoni* adult worms, which were further classified as 7936 long intergenic non-coding RNA genes (SmLINCs), i.e. lncRNAs that are transcribed from intergenic genomic loci where no protein-coding genes are transcribed, 7455 long antisense non-coding RNA genes (SmLNCAs), transcribed from the opposite strand in genomic loci where protein-coding genes are located, and 1192 long sense non-coding RNA genes (SmLNCSs), transcribed from the same strand in genomic loci of protein-coding genes. In addition, 14,520 protein-coding gene isoforms (Smps) were detected as expressed. **Table A in S1 Appendix** shows the list of all expressed lncRNA, and mRNA genes detected in the samples, along with their TPM values obtained in our re-analyses, for each gene at each of the replicate samples.

To identify the pairing-dependent lncRNAs, we compared the expression levels of lncRNA genes among the 23 different mixed-sex and single-sex adult worm samples from the 8 different conditions assayed by Lu et al., 2016 [14]. We found 3681 unique differentially expressed

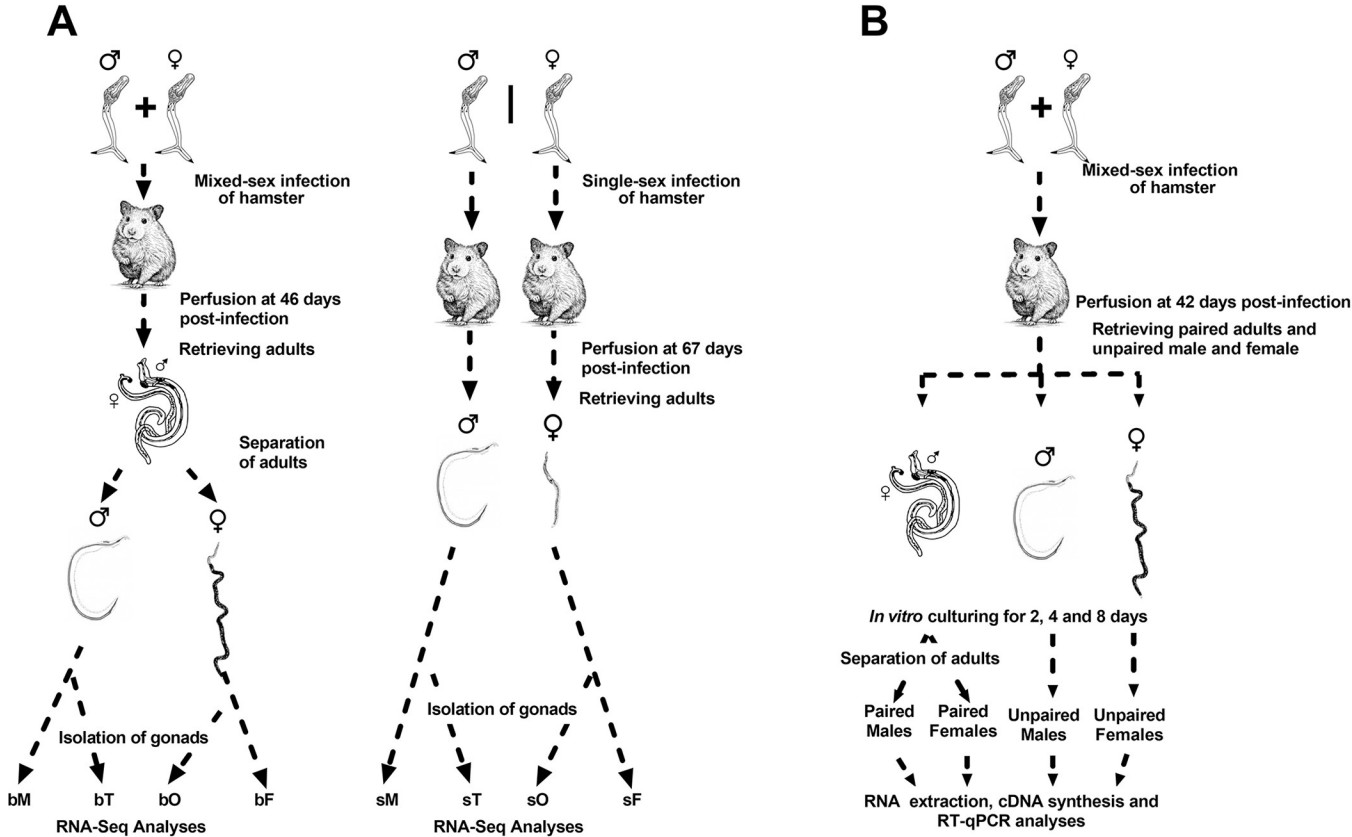

**Fig 1. Schematic representation of the pairing assays performed by Lu *et al.* 2016 and in the present work.** (**A**) Lu *et al.* [14] exposed hamsters to a *S. mansoni* mixed-sex cercariae infection (left) or to single-sex infections (middle); 46 days after infection (left) or 67 days after infection (middle), hamsters were perfused, and the adult worms recovered. For the mixed-sex infection, parasites were manually separated. Gonads were isolated from a fraction of the recovered worms, and RNA was extracted from mixed-sex infection whole males (bM) and testes (bT), from mixed-sex whole females (bF) and ovaries (bO), as well as from single-sex whole males (sM) and testes (sT) and single-sex whole females (sF) and ovaries (sO). Samples were submitted to RNA-Seq; in the present work, we re-analyzed the RNA-Seq data to search for differentially expressed lncRNAs. (**B**) Here, hamsters were exposed to a *S. mansoni* mixed-sex cercariae infection; adult worms were retrieved by perfusion 42 days post-infection. Worms retrieved in the perfusion as paired couples were collected separate from adult males and females that were retrieved as naturally separated worms. Worm pairs or separated worms were cultured *in vitro* for 2, 4 and 8 days in ABC media; separated worms are known to experience regression of the reproductive organs. At the end of the incubation times, the paired worm couples were manually separated. Male and female worms that were either paired or unpaired during *in vitro* culturing were submitted to RNA extraction, and RT-qPCR measurements were performed. Images reproduced with permission from commons.wikimedia.org (trematode lifecycle stages), from stock.adobe.com (hamsters), and from Elsevier B.V. through PLSclear ([68], immature and mature females).

(DE) lncRNA genes (**Fig 2A**), i.e. lncRNAs found as DE between mixed-sex and single-sex adult worms in at least one of the comparisons among the 23 analyzed samples (22% DE lncRNAs, out of 16,583 expressed lncRNAs). In addition, 11,109 unique protein-coding gene isoforms (out of the 14,520 isoforms, i.e. 77%) were detected as differentially expressed between mixed-sex and single-sex adult worms (**Fig 2B**).

## Selection of lincRNAs to be tested for their involvement in adult worm pairing

We selected a subset of 10 lincRNA candidates to be tested for their possible involvement in adult worm pairing, by applying a series of steps of a filtering pipeline (**Supplementary Methods in S1 Text**) that comprised the following criteria: (i) only lincRNAs were selected, because we use double-stranded RNAs to promote knockdown of their transcripts, and we wanted to avoid the problem of a simultaneous artifactual knockdown of a protein-coding message from

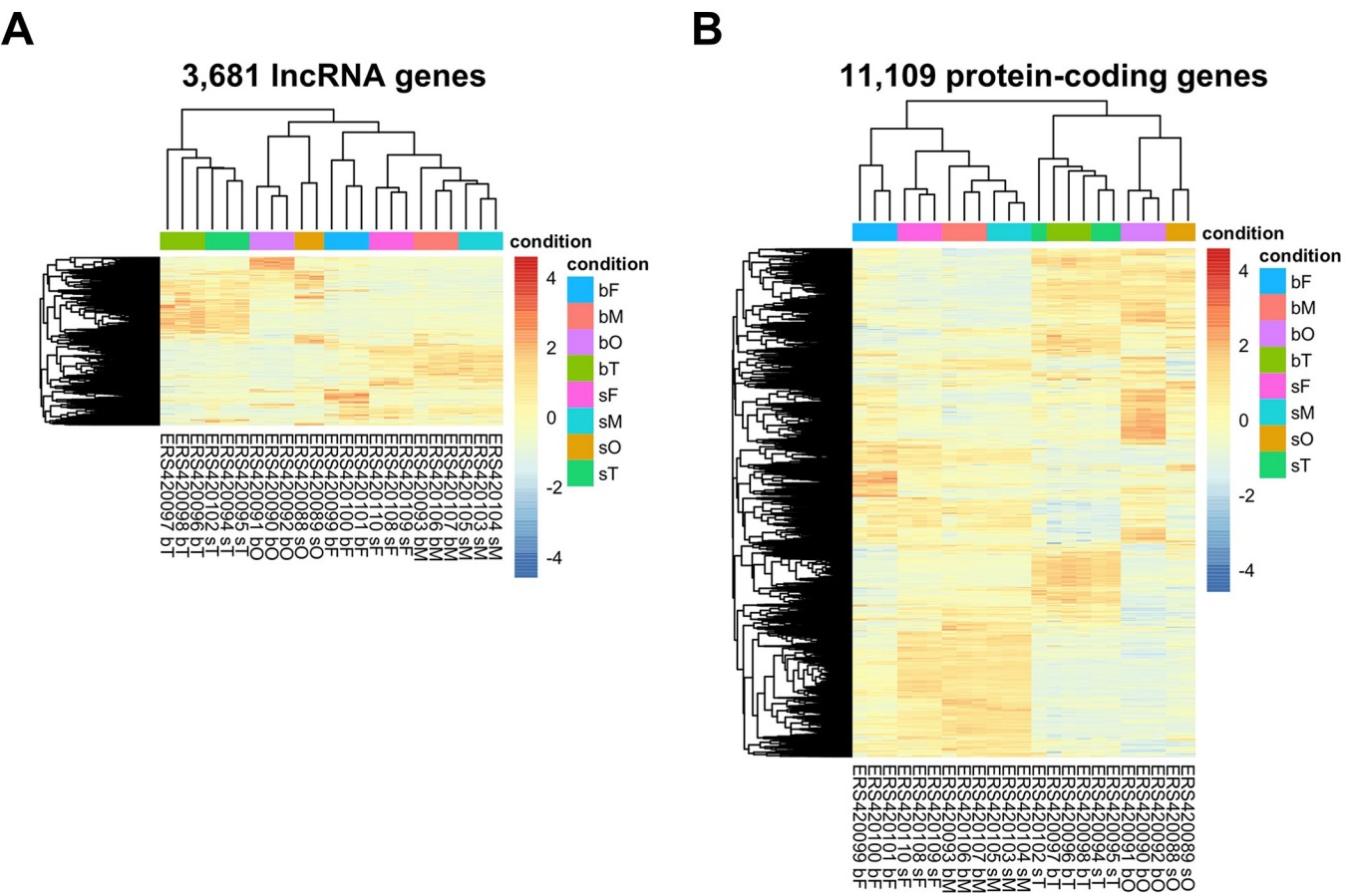

**Fig 2. Heatmap of *Schistosoma mansoni* long non-coding RNAs (lncRNAs) or protein-coding genes differentially expressed (DE) between parasites from mixed-sex and single-sex infections.** Non-supervised hierarchical clustering of (**A**) 3681 DE lncRNA genes (lines) or (**B**) 11,109 DE protein-coding gene isoforms (lines) in each of the biological replicates (columns) of *S. mansoni* parasites retrieved from mixed-sex (b) or single-sex (s) infections, as indicated in the sample labels at the bottom of the heatmaps. These results were obtained through a re-analysis of the RNA-Seq data from Lu *et al.*, 2016 [14], now using as reference the lncRNA transcriptome previously published by Maciel *et al.*, 2019 [41]. Statistically significant DE genes were determined with DESeq2 (FDR < 0.05). Gene expression levels are shown as Z-scores, which are the number of standard deviations below (blue lines, downregulated) or above (red lines, up-regulated) the mean expression value of each gene among all samples, as indicated by the color scale on the right. Females from mixed-sex or single-sex infections are identified by bF or sF; males from mixed-sex or single-sex infections by bM or sM; ovaries from mixed-sex or single-sex infection females by bO or sO; testes from mixed-sex or single-sex infection males by bT or sT.

the same locus; (ii) lincRNAs must be up-regulated in mixed-sex females, males, and their reproductive organs compared to the single-sex counterparts; (iii) we prioritized lincRNAs that were expressed at values higher than 2 transcripts per million (TPM), and that had no evidence of alternatively spliced isoforms when analyzing the full transcriptome. The final set of 10 selected lincRNAs comprises seven lincRNAs enriched in females from mixed-sex infections (bF), one lincRNA enriched in males from mixed-sex infections (bM), one lincRNA enriched in ovaries of females from mixed-sex infections (bO), and one lincRNA enriched in testes of males from mixed-sex infections (bT) (see **Supplementary Methods in S1 Text**).

## Validation by RT-qPCR of differential expression of the selected lincRNAs upon worms unpairing

To validate by RT-qPCR the pairing-dependent expression of all 10 selected lincRNAs in adult worms, we have used an *in vitro* mimetic model of paired and unpaired adult worms in which the adult worms retrieved from perfusion of hamsters infected with mixed sex cercariae were

cultured *in vitro* for up to 8 days either as paired couples or as separated worms [12] (**Fig 1B**, see also Methods). Importantly, the naturally unpaired worms retrieved from perfusion were cultured *in vitro* for mimicking the single-sex worms.

To check if this *in vitro* paired/unpaired mimetic model has characteristics similar to the mixed-sex/single-sex *in vivo* infection model used by Lu et al., 2016 [14], we first performed RT-qPCR assays to measure the expression of 14 protein-coding genes, including genes that are differentially expressed in the mixed-sex/single-sex *in vivo* infection model [14], as described in detail in the **Supplementary Data in S1 Text**. Essentially, most protein-coding genes analyzed were differentially expressed with a similar pattern both in the *in vitro* paired/ unpaired mimetic model and in the mixed-sex/single-sex *in vivo* infection model (see **Supplementary Data in S1 Text**).

Next, we evaluated the expression of all 10 selected lincRNAs using the *in vitro* paired/ unpaired mimetic model and 6 of them showed by RT-qPCR an expression pattern (**Figs 3 and 4, solid green and solid orange**) similar to that found in our re-analysis of RNA-Seq data in worms from the *in vivo* single-sex/mixed-sex infections of Lu et al., 2016 [14] (**see Figs 3 and 4, plaid green and plaid orange**).

Specifically, SmLINC142881, SmLINC175062, and SmLINC110998 were found in the RNA-Seq re-analysis to be more expressed in adult females from mixed-sex infection (bF) when compared with those from single-sex infection (sF) (**Fig 3A to 3C, plaid orange**), and they were all validated by RT-qPCR in the *in vitro* mimetic model as being more expressed in paired (P) adult females compared with unpaired (U) ones (**Fig 3A to 3C, solid orange**). Of note, SmLINC142881 showed the highest expression levels in bF (TPM average of 25, **Fig 3A, plaid orange**), while SmLINC110998 showed the largest fold-change difference between samples from paired and unpaired adult females cultured *in vitro* for 8 days (40X) (**Fig 3C, solid orange**).

SmLINC133371 had been identified as an sT, sF and sO enriched lincRNA in the RNA-Seq re-analyses (**Fig 4A, plaid green and plaid orange**), and a higher expression was confirmed by RT-qPCR in unpaired (U) adult females compared with paired (P) ones in the *in vitro* mimetic model (**Fig 4A, solid orange**).

SmLINC101519 is a lincRNA that presented a peculiar expression profile in the RNA-Seq dataset: its expression was higher in males from mixed-sex infections (bM) when compared with single-sex males (sM) (**Fig 4B, plaid green**), while an opposite pattern with higher expression in single-sex females (sF) than in females from mixed-sex infections (bF) was observed (**Fig 4B, plaid orange**). This profile was confirmed by RT-qPCR in the *in vitro* mimetic model, with adult males cultured as paired couples (P) having higher expression than unpaired adult males (U) (**Fig 4B, solid green**), whereas adult females cultured as unpaired worms (U) showed a higher expression than paired adult females (P), especially at longer culturing periods (4 and 8 days) (**Fig 4B, solid orange**).

Interestingly, SmLINC141426, which had been detected *in vivo* as enriched in testes of males from mixed-sex infections (bT) compared to testes from single-sex males (sT) (**Fig 4C, plaid green**), was validated by RT-qPCR in the *in vitro* mimetic model with whole worms as more highly expressed in paired (P) adult males when compared to unpaired (U) males (**Fig 4C, solid green**).

Four other lincRNAs have been measured by RT-qPCR and their expression profiles were not validated when comparing the *in vitro* mimetic model and the *in vivo* mixed-sex/single-sex infections. These lincRNAs are enriched in bF, bO, and sM samples in the RNA-Seq dataset (**Fig A in S1 Text**).

Taken together, these results show that a set of selected lincRNAs differentially expressed *in vivo* between worms from mixed-sex and single-sex infections are also differentially expressed

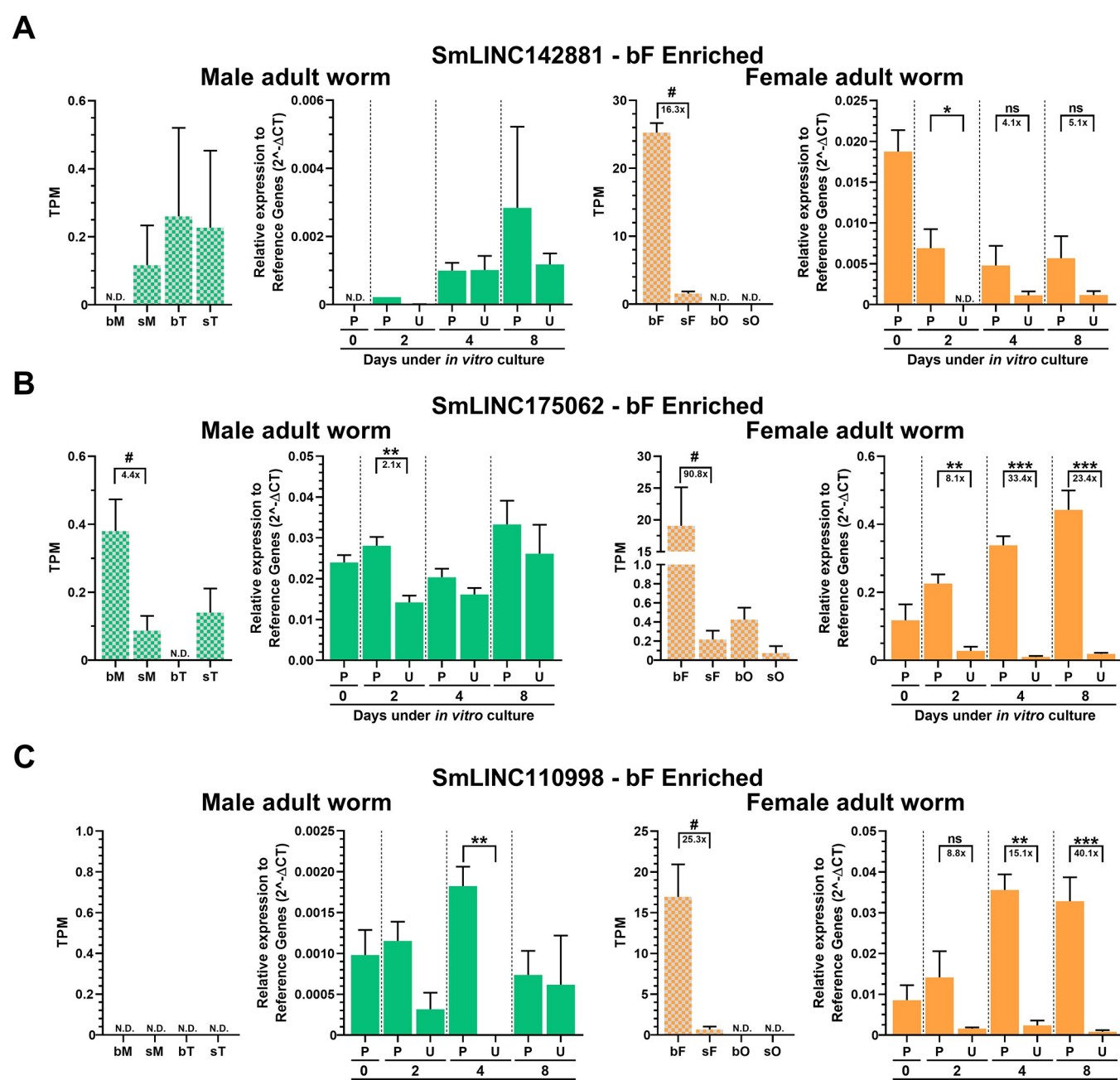

**Fig 3. Expression of lincRNAs enriched in females from mixed-sex infections, and *in vitro* differential expression validation by RT-qPCR.** Three lincRNAs detected as enriched in females from mixed-sex infections (bF) in the re-analyses of the RNA-Seq dataset of Lu et al., 2016 [14], were selected for *in vitro* RT-qPCR assays, namely (**A**) SmLINC142881, (**B**) SmLINC175062, and (**C**) SmLINC110998. Male related results are shown on the left (green) and female results on the right (orange). Paired couples (P) or unpaired (U) parasites were obtained by perfusion of hamsters infected for 42 days with *S. mansoni* cercariae. After perfusion, males and females were cultured *in vitro* for 2, 4 or 8 days as paired (P) couples or unpaired (U) male and female worms. RT-qPCR results (solid-colored graphs) are normalized to the geometric mean of reference genes Smp_099690 and Smp_023150. Expression values from 4 different biological replicates are shown. Standard error of the mean (SEM) is shown in the error bars. (*) = p < 0.05; (**) = p < 0.01; (***) = p < 0.001, Student t test. N.D.: Not detected. ns: p-value > 0.05. For comparison, RNA-Seq data from the re-analysis of Lu *et al.*, 2016 [14] is shown (plaid-colored graphs) and the expression is measured in TPM (transcripts per million); RNA-Seq data is retrieved from males (M), females (F), testes (T) or ovaries (O) from either a mixed-sex (b) or a single-sex (s) infection; (#) = FDR<0.005. The fold-change differences between the compared groups are represented under the brackets.

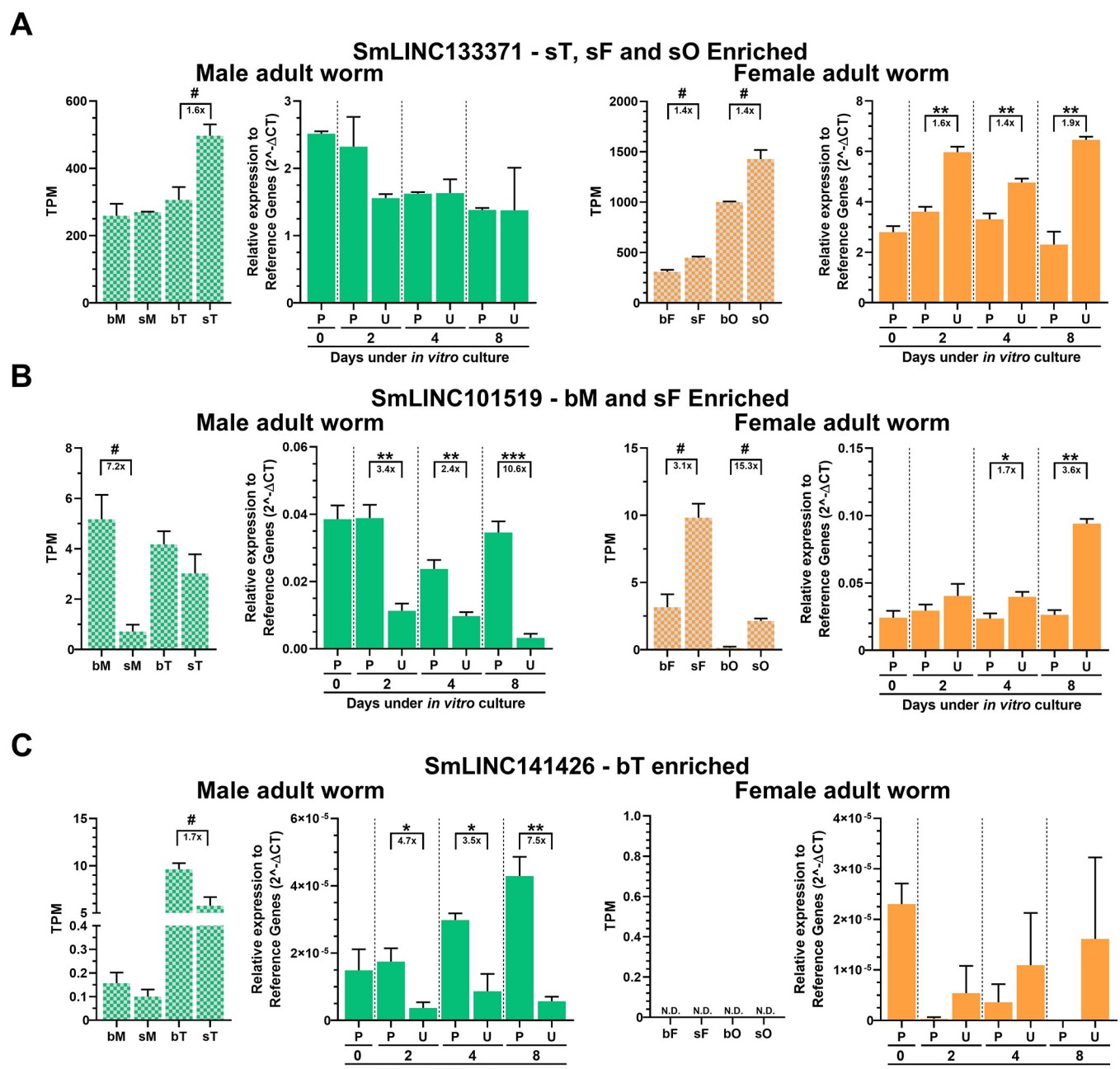

**Fig 4. Expression of lincRNAs enriched in samples other than females from mixed-sex infections and *in vitro* differential expression validation by RT-qPCR.** Three lincRNAs detected as enriched in samples other than females from mixed-sex infections in the re-analyses of the RNA-Seq dataset of Lu et al., 2016 [14], were selected for *in vitro* RT-qPCR assays, namely (**A**) SmLINC133371, enriched in sT, sF and sO; (**B**) SmLINC101519, enriched in bM and sF; and (**C**) SmLINC141426, enriched in bT. Male related results are shown on the left (green) and female results on the right (orange). Paired couples (P) or unpaired (U) parasites were obtained by perfusion of hamsters infected for 42 days with *S. mansoni* cercariae. After perfusion, males and females were cultured *in vitro* for 2, 4 or 8 days as paired (P) couples or unpaired (U) male and female worms. RT-qPCR results (solid-colored graphs) are normalized to the geometric mean of reference genes Smp_099690 and Smp_023150. Expression values from 4 different biological replicates are shown. Standard error of the mean (SEM) is shown in the error bars. (*) = $p < 0.05$; (**) = $p < 0.01$; (***) = $p < 0.001$, Student t test. N.D.: Not detected. ns: p-value > 0.05. For comparison, RNA-Seq data from the re-analysis of Lu *et* al., 2016 [14] is shown (plaid-colored graphs) and the expression is measured in TPM (transcripts per million); RNA-Seq data is retrieved from males (M), females (F), testes (T) or ovaries (O) from either a single-sex (s) or mixed-sex (b) infection; (#) = FDR<0.005. The fold-change differences between the compared groups are represented under the brackets.

upon unpairing *in vitro*, indicating potential involvement in *S. mansoni* sexual pairing and/or development.

### *In vitro* silencing of pairing-dependent lincRNAs decreases worm viability, worm pairing, oviposition and/or egg hatching

To assess the relevance of lincRNAs in worms pairing homeostasis, we performed *in vitro* silencing by soaking adult *S. mansoni* couples for eight days with double-stranded RNAs (dsRNAs) that targeted each of the lincRNAs of interest. We were able to design and synthesize dsRNAs for 3 out of the 6 pairing-dependent lincRNAs confirmed as differentially expressed in the *in vitro* mimetic model, which target SmLINC101519, SmLINC110998, or SmLINC175062; dsRNA probes were 610, 294, or 296 bases long, respectively (**Table B in S1 Appendix**).

Adults were incubated *in vitro* with each of the 3 dsRNA probes targeting the pairing-dependent lincRNAs, with a control dsmCherry (a dsRNA that will activate the RNAi pathway but will not target any parasite gene), or with no dsRNA and were followed for up to eight days. RT-qPCR assays confirmed the effective knockdown of each of the 3 lincRNAs both in males and females (**Fig 5A**), with statistically significant reductions in the lincRNAs expression levels of 70–75%, 85–90%, or 60–78% for SmLINC101519, SmLINC110998, or SmLINC175062, respectively (**Fig 5A**).

Silencing of each lincRNA caused a reduction in the pairing status of adult worms starting at day 7 and reaching a significant reduction to 10–30% pairing after 8 days in culture (**Fig 5B**). An earlier impact was observed in the adhesion of adult worms to the culture plate, starting at days 4 to 6, with a complete lack of adhesion observed at 6 to 8 days of *in vitro* culture for all silenced lincRNAs (**Fig 5C**). Upon lincRNA knockdown, motility of adult worm couples and of unpaired worms was affected at 4 to 6 days in culture, with significant reductions in motility at 8 days of silencing of 30%, 50%, or 80% for SmLINC101519, SmLINC175062, or SmLINC110998, respectively (**Fig 5D**). The viability of adult worm couples (male and females together) was assessed after 8 days of *in vitro* lincRNAs silencing by measuring the total ATP levels of parasites, and a significant 35% reduction in the viability of worms silenced for SmLINC101519 was observed (**Fig 5E**).

Reduction in adult worm fitness reflected on egg laying and egg health: egg laying was significantly reduced by 58%, 50%, or 26% in the adult worm pairs treated for 8 days with dsRNAs targeting SmLINC101519, SmLINC110998, or SmLINC175062, respectively (**Fig 5F**). Egg size was reduced upon dsRNA treatment for all lincRNAs, with a higher egg area reduction seen in eggs from worms silenced for SmLINC110998 (**Fig 5G**). Still, no difference was seen in eggshell integrity, measured by the egg autofluorescence, in any of the lincRNAs silencing conditions (**Fig 5H**). Representative images of eggs from control and silenced worms are shown in **Fig B in S1 Text**. Interestingly, egg proliferation was measured by the ratio of EdU$^+$ to DAPI$^+$ cell ratio (EdU, a thymidine analog, 5-ethynyl-2′-deoxyuridine), and a significant decrease in proliferation was observed with SmLINC101519, SmLINC110998, or SmLINC175062 silencing (**Fig 5I**). Representative images of EdU-labelled eggs from control and silenced worms are shown in **Fig B in S1 Text**. A significant reduction in egg hatching was only observed in eggs from worms treated with dsRNAs targeting SmLINC110998 or SmLINC175062 (**Fig 5J**), which indicates that silencing of different lincRNAs affected distinct pathways involved with egg development and maturation.

An unrelated control lincRNA, not detected as enriched in mixed-sex samples in the mixed-sex/single-sex model comparison, namely SmLINC130991 (**Fig C, panel A in S1 Text**) was silenced *in vitro* for 8 days. Silencing was effective, and a 50–70% reduction in the level of

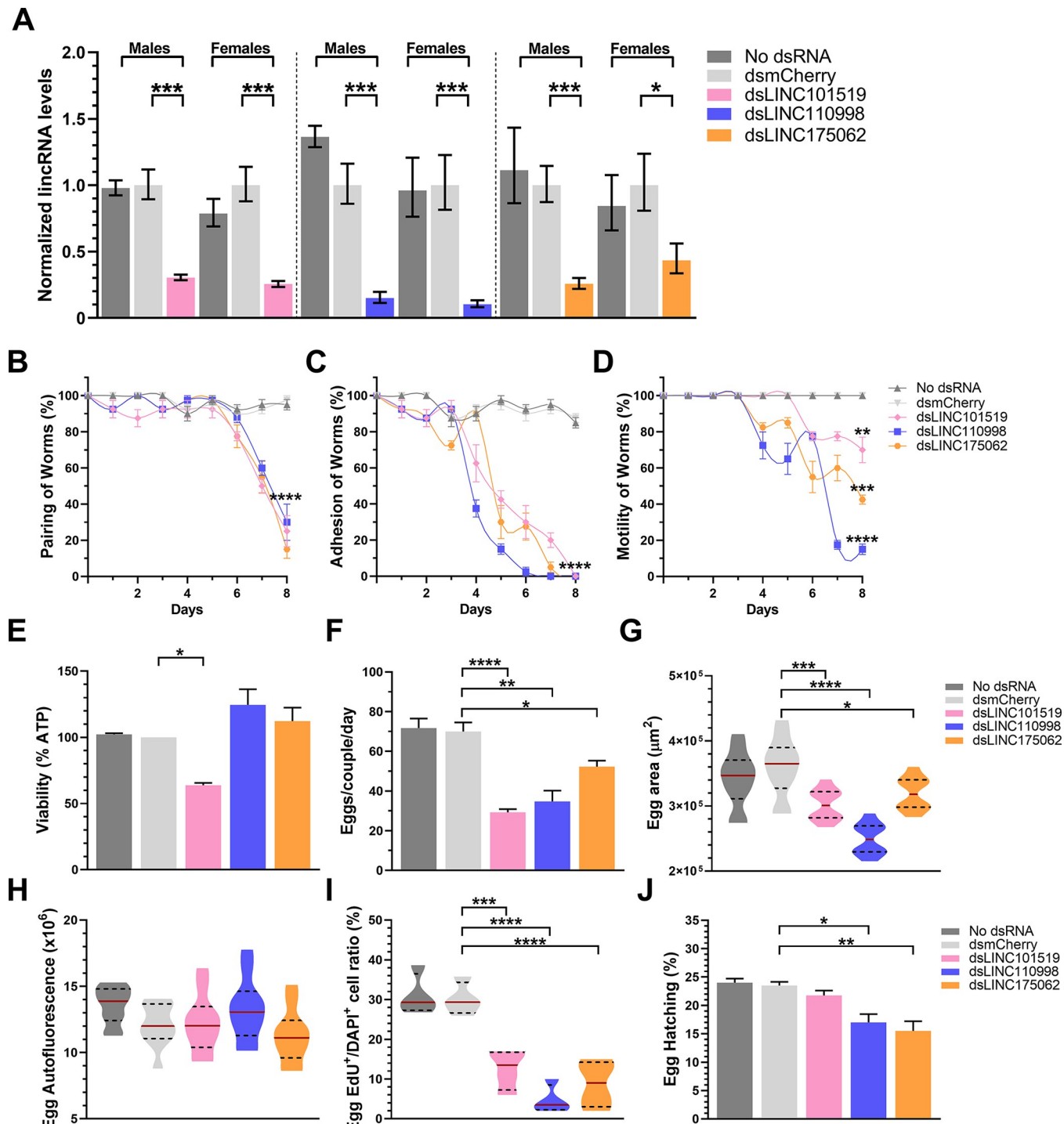

**Fig 5. Phenotypic changes in *Schistosoma mansoni* adult worm couples upon *in vitro* silencing (RNAi) of each of three pairing-dependent lincRNAs.** Paired couples were obtained by perfusion of hamsters infected for 42 days with *S. mansoni* cercariae. Couples were cultured *in vitro* for 8 days, in ABC media supplemented with 30 μg/mL of dsRNA targeting each of the lincRNAs indicated by the different colors, namely SmLINC101519 (pink), SmLINC110998 (blue), or SmLINC175062 (orange). Medium was exchanged every other day while dsRNA was added every day. dsRNA targeting mCherry (a gene that is not present in *S. mansoni*) was assayed in parallel as a negative control (light gray). Results for parasites cultured with no dsRNA are also shown (dark gray). (**A**) RT-qPCR results for each lincRNA expression level are normalized to the geometric mean of reference genes Smp_099690 and Smp_023150. (**B-D**) Pairing status, adhesion to the plate and motility of worm couples were traced along the 8 days of the experiment. (**E**) Viability of adult worms (males+females) was monitored using the ATP-Glo Assay. (**F**) At the end of the experiment (8 days), eggs were collected and counted. (**G-J**) Collected eggs were monitored for their size (area) (G), integrity of their eggshell (autofluorescence) (**H**), proliferation status of the embryos (Egg Edu+/DAPI+ cell ratio) (**I**), and the percentage of egg

hatching was measured by keeping the eggs in culture for another 7 days in ABC media for synchronization of their development, then assessing egg hatching as described in Methods, with the percentage of hatched eggs being shown (**J**). Violin plot representation at Figs (**G-I**) with the median indicated by the red line and the quartiles represented by the dashed lines. Results from 4 different biological replicates are shown. Standard error of the mean (SEM) is shown in the error bars. Student t test (panels A to D) or One-Way ANOVA test with multiple comparisons to dsmCherry group (panels E to J) were used. (*) = p < 0.05; (**) = p < 0.01; (***) = p < 0.001; (****) = p < 0.0001.

SmLINC130991 was obtained (**Fig C, panel B in** S1 Text). Yet, none of the phenotypic changes that were observed with silencing of the three selected pairing-dependent lincRNAs was present upon silencing of SmLINC130991 (**Fig C, panels C to K in** S1 Text).

## Cell proliferation status and vitellaria composition of adult worms are affected by the i*n vitro* silencing of pairing-dependent lincRNAs

We also investigated the proliferation status of adult worms (**Fig 6A**) and their gonads (**Fig 6B**) upon silencing of the pairing-dependent lincRNAs. This was achieved by pulse chasing with EdU for 24 hours. While mCherry dsRNA treatment did not affect the adult worm nor their gonads proliferation, SmLINC101519 silencing impacted the adult male body cells proliferation status (**Fig 6A, left**), but neither the female body cells (**Fig 6A, right**) nor the male and female gonads were affected (**Fig 6B**). Cell proliferation was quantified by measuring the number of EdU$^+$ cells per $\mu m^2$ (**Fig D in** S1 Text). Upon SmLINC101519 silencing, the median number of EdU$^+$ cells per $\mu m^2$ in adult male head and body reduced from 1.8 to 0.02 and in adult male tail from 1.2 to 0.02 (**Fig D, panel A in** S1 Text) whereas no change was observed in adult females, in testes or in ovaries.

In contrast, SmLINC110998 silencing markedly reduced the female ovary cells proliferation (**Fig 6B, right** and **Fig D, panel D in** S1 Text) and the female body cells proliferation (**Fig 6A, right**) as shown by the reduced number of EdU$^+$ cells per $\mu m^2$ in the female head, body, and tail (**Fig D, panel B in** S1 Text), with no detectable effect in males. These results are in agreement with the fact that SmLINC110998 is highly expressed in females, being barely detected in males (Fig 3C).

As for silencing of SmLINC175062, we have observed a decrease only in male adult worm body cells proliferation (**Fig 6A, left**) with a significant decrease in the number of EdU$^+$ cells per $\mu m^2$ in adult male head, body, and tail (**Fig D, panel A in** S1 Text), whilst female adult worms (**Fig 6A, right** and **Fig D, panel B in** S1 Text) and the male and female gonads proliferation were not affected (**Fig 6B** and **Fig D, panels C and D in** S1 Text).

Female vitellaria composition was evaluated by double staining of vitelline and lipid droplets within the vitellaria, as previously described [30]. Upon SmLINC110998 silencing the female vitellaria was impacted mainly at the vitelline droplets (pink staining) (**Fig 7**), corroborating with the egg-laying reduction (**Fig 5F**). We quantified the extent of vitelline droplets and lipid droplets in the vitellaria, as measured by Relative Fluorescence Units (RFU) per $\mu m^2$ (**Fig E in** S1 Text). SmLINC101519 silencing caused a 75% reduction in the median RFU signal of vitelline droplets (**Fig E, panel A in** S1 Text) and a smaller, however significant reduction of 35% in the median RFU signal of lipid droplets (**Fig E, panel B in** S1 Text). SmLINC110998 silencing caused a decrease in vitelline droplets (**Fig 7**) with marked reduction of 92% in its median RFU signal (**Fig E, panel A in** S1 Text), with no detectable effect on lipid droplets. SmLINC175062 silencing slightly decreased lipid droplets only (**Fig E, panel B in** S1 Text). Of note, these assays demonstrated that dsmCherry control treatment did not disturb the vitellaria of adult worm females (**Fig E, panels A and B in** S1 Text).

Silencing of the unrelated control SmLINC130991 did not affect the EdU labelling of male or female body cells, of ovaries or testes (**Fig C, panel L, and Fig D, panels A to D in** S1 Text), or the labelling of lipid and vitelline droplets (**Fig C, panel M, Fig E panel A and B in** S1 Text).

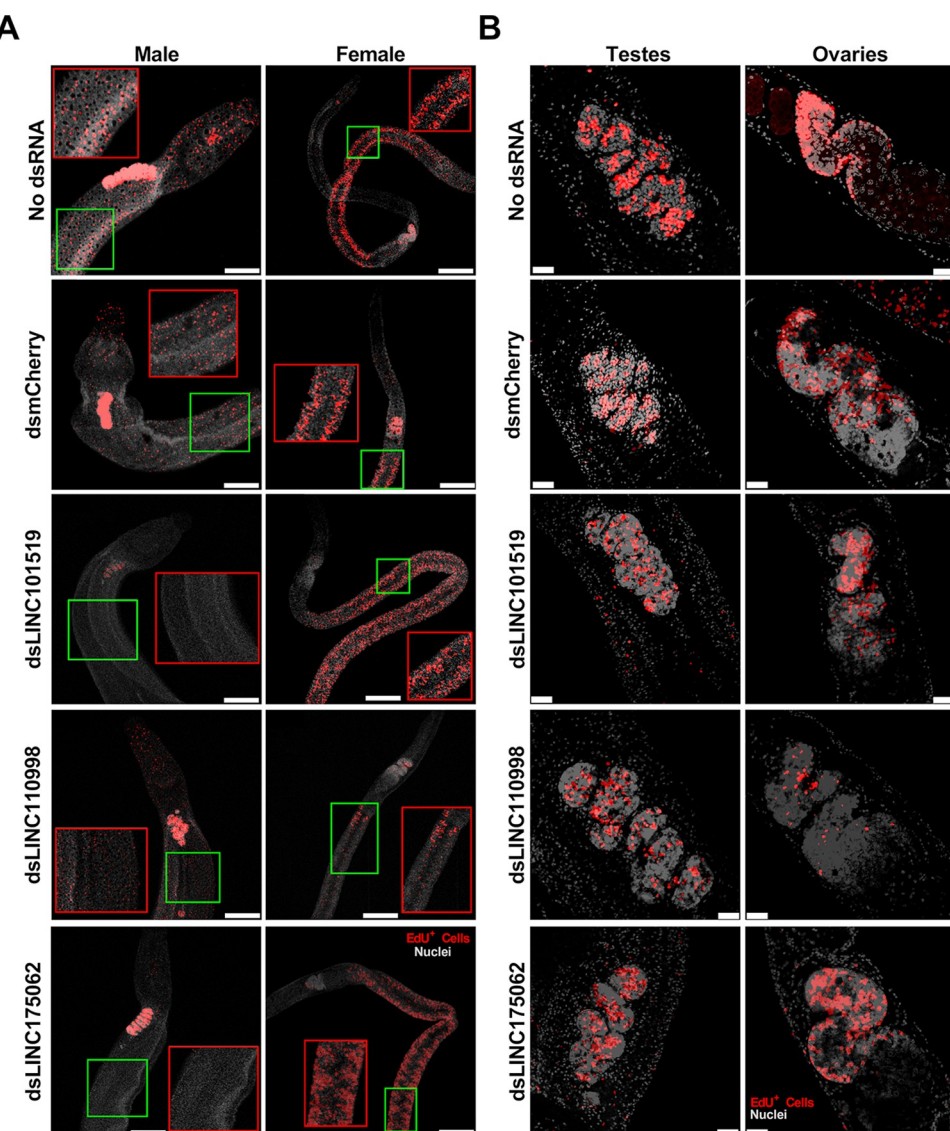

**Fig 6.** ***In vitro*** **silencing of pairing-dependent lincRNAs in** ***S. mansoni*** **adult worm couples leads to an impaired cell proliferation status.** Paired couples were obtained by perfusion of hamsters infected for 42 days with *S. mansoni* cercariae. Couples were cultured *in vitro* for 8 days in ABC media supplemented with 30 µg/mL of dsRNA targeting each of the indicated lincRNAs, namely SmLINC101519, SmLINC110998, or SmLINC175062, with a negative control dsRNA targeting mCherry (a gene that is not present in *S. mansoni*) or with no dsRNA. Medium was exchanged every other day while dsRNA was added every day. Cell proliferation was assayed by labeling with thymidine analog 5-ethynyl-2′-deoxyuridine (EdU), which was added to the cultures on the 7[th] day of culture at a final concentration of 10 µM and incubating for 24 h. EdU labeling detection in (**A**) adult worms, and in (**B**) the gonads was performed as described [64]. DAPI stained cells nuclei are shown in gray and EdU+ cells (proliferating cells) are stained in red. Scale bars: 250 µm for the adult worm images in (A), and 25 µm for the adult worm gonad images in (B). Representative images from 3 experiments with n > 10 parasites. The red rectangles define zoomed-in insets of interest that correspond to the regions within green rectangles.

## LincRNAs involved in pairing are expressed in worm reproductive tissues

To gain further insight into the possible functions of the selected pairing-dependent lincRNAs, we performed whole mount *in situ* hybridization (WISH) with adult female and male worms for four of the six lincRNAs validated by RT-qPCR (**Fig 8** and **Fig F in S1 Text**).

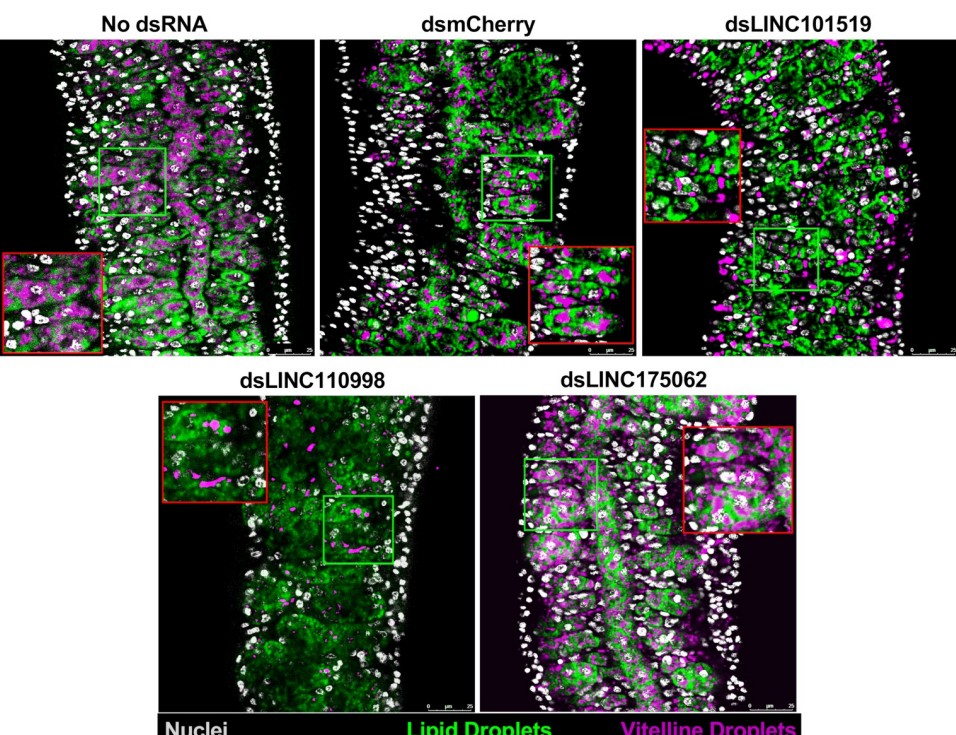

**Fig 7. *In vitro* silencing of pairing-dependent lincRNAs in *S. mansoni* adult worm couples causes female vitellaria impairment.** Paired couples were obtained by perfusion of hamsters infected for 42 days with *S. mansoni* cercariae. Couples were cultured *in vitro* for 8 days in ABC media supplemented with 30 μg/mL of dsRNA targeting each of the indicated lincRNAs, namely SmLINC101519, SmLINC110998, or SmLINC175062. Medium was exchanged every other day while dsRNA was added every day. dsRNA targeting mCherry (a gene that is not present in *S. mansoni*) was assayed in parallel as a negative control. Results for parasites cultured with no dsRNA are also reported. Female vitellaria were stained with Fast Blue BB (pink) and BODIPY (green), which labeled vitelline and lipid droplets in the vitellaria, respectively. DAPI staining of cells nuclei is shown in gray. Scale bars: 25 μm. Representative images from 3 experiments with n > 10 parasites. The red squares define zoomed-in insets of interest that correspond to the regions within green squares.

SmLINC101519 was expressed in the female uterus, vitellaria and around the female ovary, while in males, SmLINC101519 was present in testes, oesophagus and along the head and trunk (**Fig 8A, right).** Single-cell RNA-Seq (scRNA-Seq) analysis of lincRNAs expression [28] showed that SmLINC101519 was detected in parenchyma, muscle and neuron cell clusters (**Fig 8A, left, cells clusters marked with 1, 2 and 3**). Combining the WISH localization (i.e. around the female ovary) and the scRNA-Seq localization (at a number of muscle cell populations), we speculate that this lincRNA might not be expressed in oocytes but rather in the muscle cells surrounding the ovary as well as in the thick muscle layer known to be present in the uterus [31].

SmLINC110998 showed no relevant WISH signal in males, while in females the expression was vastly found in mature vitellocytes in both the vitellaria and the vitelline duct (**Fig 8B, right**), which was consistent with the scRNA-Seq analysis in which SmLINC110998 was detected in the late vitellocytes cluster (**Fig 8B, left, cells cluster 4**).

Finally, SmLINC175062 *in situ* hybridization did not detect expression in male worms, while in female worms the expression appeared to localize in mature vitellocytes (**Fig 8C, right)**; SmLINC175062 was poorly detected by scRNA-Seq, with little to no expression in any of the cell clusters (**Fig 8C, left**).

An additional pairing-dependent lincRNA, which was not assayed for silencing, was localized by WISH in the reproductive tissues of the parasite; **Fig F in S1 Text** shows that

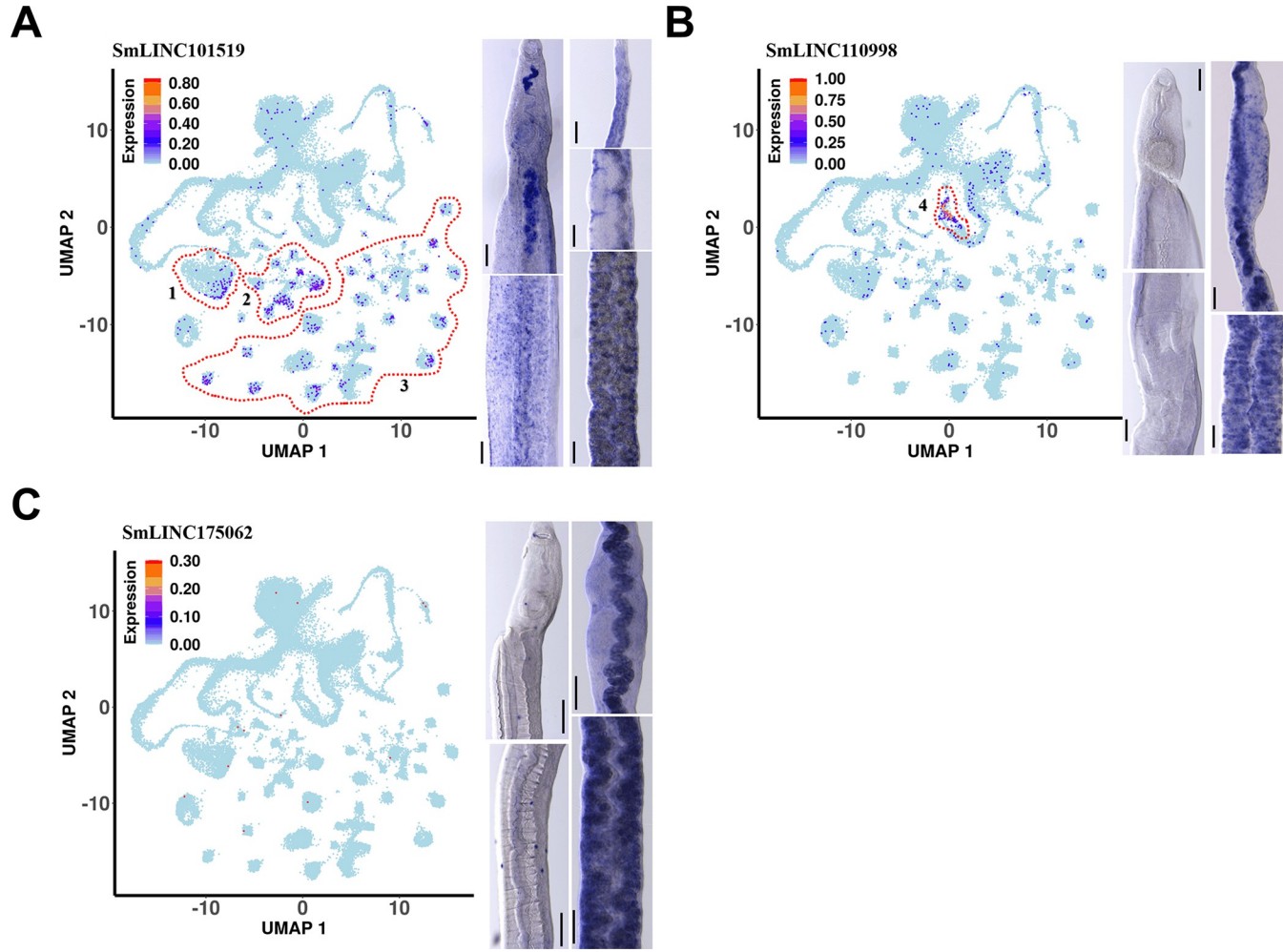

**Fig 8. Localization of the selected pairing-dependent lincRNAs in adult worm tissues by whole mount *in situ* hybridization.** Whole mount *in situ* hybridization (WISH) of each lincRNA is shown as the blue color stains in the male (left) or female (right) adult worm images of the heads and bodies. Scale bars are 100 μm. Results for (**A**) SmLINC101519, (**B**) SmLINC110998, and (**C**) SmLINC175062. For comparison purposes, single-cell RNA-Seq data from Morales-Vicente et al., 2022 [28] was retrieved from (http://verjolab.usp.br:8081/). LincRNA expression patterns across the single-cell clusters are shown with UMAP plots, colored by gene expression levels (blue = low, red = high), and the scale represents log10(UMIs+1). The regions enclosed by red dashed lines indicate the relevant cell clusters: Region 1, parenchyma 1 cells cluster; Region 2, different muscle cell clusters; Region 3, different neuron cell clusters; Region 4, late vitellocyte cells cluster.

SmLINC141426 was not detected by WISH in the male, whereas in the female, signals were detected along the head, ovary, and mature vitellocytes; with scRNA-Seq, SmLINC141426 was poorly detected in a few cells of the male gametes cell cluster (**Fig F in S1 Text**).

Finally, the unrelated control SmLINC130991 was assessed by WISH. Weak signals were observed in the female ovary and vitellarium as well as in the male testes and dispersed in the trunk (**Fig C, panel N in S1 Text**). SmLINC130991 was detected by scRNA-Seq at very low levels of expression in a few cells, not being enriched in any cell cluster (**Fig C, panel N, left, in S1 Text**).

### *In vivo* silencing of pairing-dependent lincRNAs reduces worm survival

We further checked if the *in vivo* knockdown of pairing-dependent lincRNAs would cause phenotypic effects on the parasites, as was observed *in vitro*. *In vivo* silencing approaches have

been used with *Schistosoma* in the past, using small interfering RNAs [32] or dsRNAs [33]. For dsRNAs, Li et al. [33] tested the intravenous injection in mice of 10–30 μg dsRNA per dose, and recommended at least four doses for the efficient silencing of *S. japonicum* protein-coding genes [33].

Based on the above information we decided to *in vivo* silence the pairing-dependent lincRNAs by injecting *S. mansoni*-infected mice at 7, 21, 35, and 42 dpi with 30 μg dsRNA per dose, targeting each of the three lincRNAs. On the 49th dpi, mice were perfused, and worms were recovered. RT-qPCR assays confirmed the effective knockdown of the lincRNAs in the recovered male or female worms, with reductions in the lincRNAs expression levels of 59–77%, 60–64%, or 70–78% for SmLINC101519, SmLINC110998, or SmLINC175062, respectively (**Fig 9A**).

Most interesting, the *in vivo* silencing of each lincRNA significantly decreased the number of adult worm couples recovered from the infected mice, with reductions of 55%, 57%, or 42% for SmLINC101519, SmLINC110998, or SmLINC175062, respectively (**Fig 9B**). Although no differences were found in the number of naturally unpaired males (**Fig 9C**) and females (**Fig 9D**) recovered in the perfusion in any of the silencing conditions tested, the total number of recovered worms (**Fig 9E**) decreased significantly by 31%, 35%, or 26% for SmLINC101519, SmLINC110998, or SmLINC175062 silencing, respectively.

Surprisingly, no difference in the number of eggs per gram of liver was observed (**Fig 9F**), nor a change in the egg size (**Fig 9G**) or eggshell integrity (**Fig 9H**) in the eggs recovered from the livers of infected mice. This suggests that the worm-couples that died upon the *in vivo* silencing of the pairing-dependent lincRNAs had probably ceased laying eggs only at a late point in time in our experimental setup, near the day of perfusion on the 7th week post-infection. By this time, the eggs released by sexually mature females [34] during weeks 5 to 7 would have already accumulated in the mice livers, thus masking the eventual impact of the decreased worm burden on the number of eggs per gram of liver detected after perfusion on week 7.

Most importantly, a significant 34% or 40% decrease in egg hatching was observed in the eggs from mice that were treated with dsRNAs targeting SmLINC110998 or SmLINC175062, respectively (**Fig 9I**). These were the same lincRNAs whose *in vitro* silencing caused a decrease in egg hatching (**see Fig 5J**), again indicating that silencing of these particular lincRNAs affected distinct pathways involved with egg development and maturation.

Taken together, the above results show that *S. mansoni* lincRNAs can be silenced *in vitro* and *in vivo*. In summary, this is the first evidence in *S. mansoni* of different pairing related lincRNAs being involved with some critical biological events of the parasite such as cell proliferation, female vitellaria development, or female reproduction.

## Discussion

Here, we show that long intergenic non-coding RNAs are differentially expressed between *S. mansoni* paired and unpaired adult worms cultured *in vitro*. Further, we provide a proof of concept that *in vitro* and *in vivo* silencing of selected pairing-dependent lincRNAs can cause a reduction of adult worm pairing or worm burden, of worm viability, of reproductive organ proliferation, and of egg laying or egg hatching. The *in vitro* unpairing mimetic model used here can provide a good tool for understanding the pairing-dependent homeostasis of the adult worms, and for further exploring the detailed mechanisms of action of lincRNAs, since these worms develop to their fullest potential before being unpaired *in vitro*. Of note, we have selected only fully developed, mature parasites that were collected as unpaired worms in the perfusions, as described in the Methods. Nevertheless, we cannot determine for how many days those worms were already separated in the hamster, or whether they freshly separated

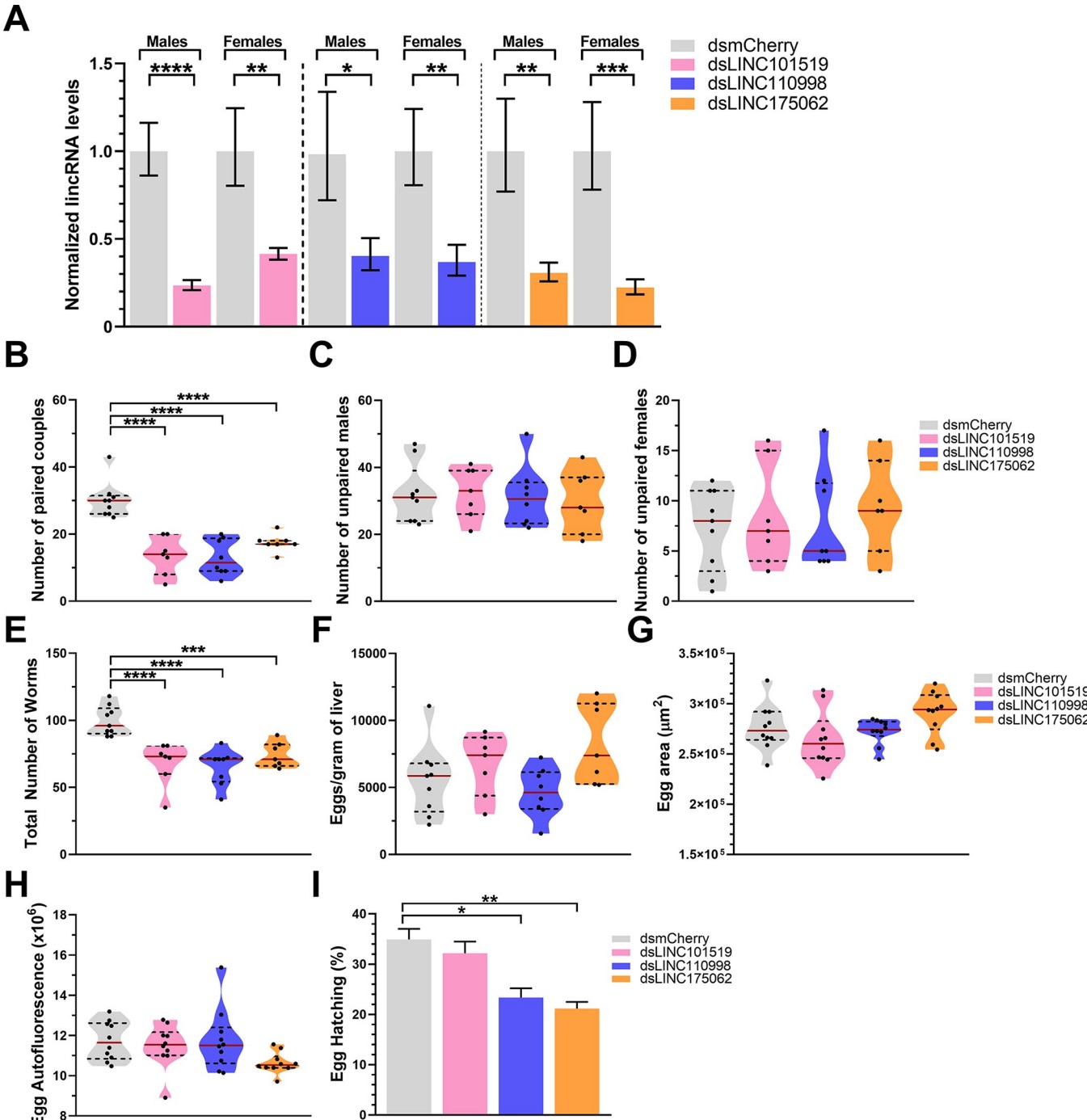

**Fig 9. *In vivo* silencing (RNAi) of pairing-dependent lincRNAs decreases the number of *S. mansoni* adult worms recovered from infected mice.** Five weeks old C57BL/6 female mice were infected with an average of 145 cercariae. At weeks 1, 3, 5 and 6 post infection, mice were injected in the retro orbital vein with 30 μg of dsRNA targeting the specified lincRNA (in a 100 μL solution with non-pyrogenic saline), namely SmLINC101519 (pink), SmLINC110998 (blue), or SmLINC175062 (orange). A dsRNA targeting mCherry (a gene that is not present in *S. mansoni* and mice) was assayed in parallel as a negative control. At week 7, the mice were perfused, and the worms retrieved. The liver from each mouse was processed for measuring eggs. (**A**) RT-qPCR results for each lincRNA expression level are normalized to the geometric mean of reference genes Smp_099690, Smp_023150, Smp_196510, and Smp_101310. (**B-E**) Worm burden results are reported, where the number of paired worm couples (**B**), number of unpaired male worms (**C**), number of unpaired female worms (**D**) retrieved from perfusion, and total number of worms (**E**) are shown. The number of eggs retrieved from the livers of infected mice was measured after liver processing as mentioned in Methods. Those eggs were counted and the number of eggs per gram of liver is reported (**F**). Egg health parameters such as size (area) (**G**) and eggshell integrity (autofluorescence) (**H**) were measured. The eggs were kept in culture for another 7 days in ABC media for synchronization of their development. Then egg hatching assay was performed as described in Methods and the percentage of hatched eggs is shown (**I**). Standard error of the mean

(SEM) is shown in the error bars. Student t test (panel A) or One-Way ANOVA test with multiple comparisons to dsmCherry group (panels B to I) were used. (*) = p < 0.05; (**) = p < 0.01; (***) = p < 0.001; (****) = p < 0.0001.

during perfusion. Since the variance of gene expression between replicates that we obtained was quite reasonable, we argue that the eventually diverse differentiation level of parasites that enter the same experimental group did not have an impact on the assays. On the other hand, the mixed-sex/single-sex cercariae infection of mammalian host model [14] is a more drastic approach for understanding the role of pairing in maintaining homeostasis in these parasites and may also reflect the impact of single sex infection on the developmental trajectory of the parasites.

Hundreds of lncRNAs were found to be differentially expressed among the RNA-Seq samples of the mixed-sex/single-sex cercariae infections of the mammalian host, in the dataset generated by Lu et al., 2016 [14]. While short RNAs (especially miRNAs) have been more explored in helminths, lncRNAs have received little attention, with the most characterized organism being schistosomes [27]. A previous work by our group has shown a lincRNA, that is up-regulated upon 5-Azacytidine treatment, to be critical for biological events of the parasite such as cell proliferation, female vitellaria development and female reproduction [27]. Yet, no other attempt to functionally characterize the roles of lncRNAs in *S. mansoni* has been made.

A detailed investigation on the male:female pairing process has been recently published, in which Chen and collaborators showed that this process induces the GLI1-dependent expression of a non-ribosomal peptide synthetase in male worms, culminating in the production of a dipeptide formed of β-alanyl-tryptamine (BATT) [20]. The BATT molecule is released from males to act as a pheromone in females, stimulating its sexual development [20]. It remains to be tested if any lincRNA would be involved in BATT signaling.

Taken together, our results show that lincRNAs are important for adult worm pairing, viability, and/or fertility *in vitro*. Silencing of some lincRNAs such as SmLINC110998 and SmLINC175062 did not affect viability (Fig 5E) while affecting motility (Fig 5D). Therefore, we cannot rule out the possibility that aberrant motility also plays a role in reducing worm pairing. In contrast, knockdown of SmLINC101519 caused a significant 40% reduction in worms' viability (Fig 5E), with reduced egg cells proliferation (Fig 5I) and sustained female somatic stem cells proliferation (Fig D in S1 Text). This suggests that, when parasite homeostasis was affected by knockdown of SmLINC101519, egg and body cell proliferation might be the sources of highest ATP consumption, as opposed to worm motility. In fact, knockdown of SmLINC101519 caused the least effect on motility (Fig 5D). On the other hand, motility was reduced by 60 to 90% upon knockdown of the other two lincRNAs (SmLINC110098 and SmLINC175062), which was not accompanied by reduction in viability (Fig 5E), again suggesting that the effects on motility were not directly correlated with ATP levels.

There is a complex interplay between the different phenotypic parameters that were evaluated here, in determining the trajectory of sustained fertility and reproduction in paired worms. It is apparent that silencing each of three different lincRNAs in paired couples kept in culture for 8 days has affected worms' homeostasis to different extents and with different time courses. Worms' adhesion and motility started falling after 4 days of silencing (Fig 5C and 5D), which may have impacted the ability of couples to remain paired. Pairing has fallen to 50% after 7 days of silencing and on the 8th day only 10 to 30% couple pairs remained among the three different dsLincRNA-silenced worms (Fig 5B). While physical separation of females from their male partners will play a role in reducing egg production and cell proliferation, our data suggest that reduced pairing is not the single drive for altered reproduction and that different lincRNAs silencing have altered reproduction homeostasis in different ways. Thus,

silencing of SmLINC175062 caused 85% couple pairing reduction at the 8[th] day (Fig 5B) with a significant, however the smallest reduction in egg production of 25% (Fig 5E) and no reduction in vitelline droplets (Fig E, panel A in S1Text), while the eggs deposited by SmLINC175062-silenced females had the highest reduction in hatching (35%) (Fig 5J). In contrast, silencing of SmLINC101519 caused 75% couple pairing reduction (Fig 5B) with the highest reduction in egg production of 60% (Fig 5E) and a significant reduction of 75% in vitelline droplets (Fig E, panel A in S1Text), while the eggs deposited by SmLINC101519-silenced females had no reduction in hatching ability (Fig 5J). Further identification of molecular mechanisms of action of these lincRNAs is warranted.

Of note, there appears to be a sex dependent differential effect of lincRNAs silencing on proliferative phenotype parameters. SmLINC101519 was the only lincRNA selected for silencing that was enriched in mixed-sex males (bM) compared with single-sex males (sM), and the opposite, enriched in single-sex females (sF) compared with mixed-sex females (bF) (Fig 4C), being expressed at levels > 2 TPM in both males and females. We measured body cell proliferation separately in each sex, and there was a clear sex difference in this phenotype. A marked decrease was observed in male body cell proliferation in dsLINC101519 treated worms (Fig D, panel A in S1 Text), while no decrease was seen in females (Fig D, panel B in S1 Text). Silencing of SmLINC175062 also caused a similar decrease in male body cells proliferation (Fig D, panel A in S1 Text), and no change in females. On the contrary, silencing of SmLINC110998 caused no change in males body cell proliferation (Fig D, panel A in S1 Text), while leading in females to a marked decrease in body cell proliferation (Fig D, panel B in S1 Text) and a decrease in vitelline droplets (Fig E, panel A in S1 Text). It is apparent that these lincRNAs interfere in different ways, either directly or indirectly with proliferative and/or reproductive phenotypes, and identification of the molecular mechanisms that are involved should be pursued.

SmLINC101519 is an 882-nt long lincRNA with two exons and an intron size of 3875 nt, mapping to chromosome 1 (SM_V7_1:11,062,339–11,067,098). *In vitro* silencing of SmLINC101519 has led to a major decrease in female oviposition, egg cell proliferation, and male adult worm cell proliferation, not affecting female body cell proliferation. On chr 1, two protein-coding genes are flanking the genomic region of SmLINC101519, namely T-box transcription factor TBX20 (Smp_003900) upstream, and High-affinity cGMP-specific 3',5'-cyclic phosphodiesterase 9A (PDE9A) (Smp_342020) downstream of the SmLINC101519 locus (see locus in the genome browser). Inspection of the expression profiles of the two protein-coding genes and the SmLINC101519 in the adult worms scRNA-Seq data [28] **(Fig G, panel A in S1 Text)** shows that both Smp_003900 and SmLINC101519 are expressed in the same cell clusters, namely muscle 1 and neuron 17 **(Fig G, panel A in S1 Text)**; one could raise the hypothesis that SmLINC101519 could be *cis*-acting to activate the expression of the neighbor TBX20 protein-coding gene, or that both TBX20 Smp_003900 and the SmLINC101519 are simultaneously co-regulated; however, further direct experimentation is needed to test this hypothesis. The major phenotypes observed upon SmLINC101519 silencing are in direct agreement with its expression profile and neighboring genes. TBX20 (Smp_003900) is essential for motor neuron development in invertebrates [35] and SmLINC101519 silencing impacted adult worm adhesion and pairing by 75–100%. While PDE9A plays important roles in cell proliferation regulation in different parasites [36, 37].

SmLINC110998 is a 322-nt long lincRNA with two exons and an intron size of 1025 nt, mapping to chr 1 (SM_V7_1:54,991,086–54,992,435). *In vitro* silencing of SmLINC11099 impacted female oviposition, egg cell proliferation, female gonad, and adult worm cell proliferation status, and female vitellaria maintenance. Those phenotypes are in concordance with the expression profile of SmLINC110998 in females, which is highly detected in the late

vitellocytes cell cluster [28] (**Fig 8B, cluster 4**). Schistosomes produce eggs consisting of an oocyte surrounded by specialized "yolk" cells known as vitellocytes [38], which provide both nutrition for the developing zygote and constituents essential for the construction of the egg shell [30]. Vitellocytes are produced by the vitellarium, which is composed of a network of thousands of follicles in which S1 stem cells differentiate to ultimately produce mature vitellocytes (S4 cells). These mature vitellocytes are fed anteriorly through the vitelline duct and are joined with fertilized oocytes in the ootype where the mature egg is formed. Disruption of mature vitellocytes production has not impacted cell proliferation status but indeed affected egg laying and egg development [30]. Semaphorin-5A (Smp_159050) and Lysyl oxidase homolog 2B (Smp_159060) are the genomic upstream and downstream neighbors of SmLINC110998, respectively; however, neither of these protein-coding genes has an expression pattern that is similar to the cell clusters' expression pattern of SmLINC110998 (**Fig G, panel B in S1 Text**), thus providing no clues to a possible regulation *in cis* eventually exerted by the lincRNA.

Finally, SmLINC175062 is a 669-nt long transcript with three exons mapping to chromosomes ZW (SM_V7_ZW:69,884,648–69,886,379) with Smp_096310 Protein kinase C zeta type (PKCZ) and Smp_096290 Transmembrane protein 256 homolog genes as its upstream and downstream neighbor genes (**Fig G, panel C in S1 Text**); because SmLINC175062 was not detected by scRNA-Seq, there is no evidence of a possible effect of the lincRNA on the neighbor protein-coding genes. SmLINC175062 silencing caused major effects in female egg laying and egg proliferation, with a male adult worm cell proliferation defect as well. SmLINC175062 expression was only observed by WISH in females, spread across their mature vitellocytes (**Fig 7C**), which is consistent with the female and egg phenotypes observed. Interestingly, SmLINC175062 silencing caused no decrease in vitelline droplets (Fig E, panel A in **S1 Text**) while a significant decrease in lipid droplets was observed (Fig E, panel B in **S1 Text**), suggesting a possible role of this lincRNA on the vitellaria lipid metabolism.

In summary, our *in vitro* and *in vivo* silencing experiments have shown that lncRNAs play pivotal roles at different aspects involved with homeostasis maintenance of schistosome adult worms. To our knowledge, this is the first report demonstrating the *in vivo* silencing of lncRNAs in parasites. We obtained evidence of major phenotypes upon *in vivo* silencing of selected lincRNAs, such as reduced worm burden and decreased egg hatching; further detailed characterization of the different molecular mechanisms of action that were affected is warranted. In addition, we provide an extensive repository of lincRNAs differentially expressed between unpaired and paired worms, which is an important asset for future exploration of *S. mansoni* lincRNAs as possible *in vivo* targets against schistosomiasis.

## Methods

### Ethics statement

The experimental protocols were in accordance with the Ethical Principles in Animal Research adopted by the Conselho Nacional de Controle da Experimentação Animal (CONCEA) and the protocol/experiments have been approved by the Comissão de Ética no Uso de Animais do Instituto Butantan (CEUAIB number 8859090919). This study was carried out in compliance with the ARRIVE (Animal Research: Reporting of In Vivo Experiments) guidelines (https://arriveguidelines.org/arrive-guidelines).

### Analysis of RNA-Seq data

Public RNA-Seq data from Lu et al. [14] for *S. mansoni* males, females and their gonads were downloaded from the SRA-NCBI database (project number PRJEB1237; bM: #ERS420093,

#ERS420106, #ERS420107; sM: #ERS420103, #ERS420104, #ERS420105; bF: #ERS420099, #ERS420100, #ERS420101; sF: #ERS420108, #ERS420109, #ERS420110; bT: #ERS420096, #ERS420097, #ERS420098; sT: #ERS420094, #ERS420095, #ERS420102; bO: #ERS420090, #ERS420091, #ERS420092; sO: #ERS420088, #ERS420089). Adapters and bad quality reads were filtered out using fastp v. 0.19.5 with default parameters [39]. For transcripts expression quantitation the genome sequence v.7, and a GTF file containing the protein-coding transcriptome v 7.1 were downloaded from the WormBase ParaSite resource (version WBPS14) [40]. The latter was merged with the lncRNA transcriptome sequences identified by Maciel et al. [41] and the resulting GTF, which is available at http://verjolab.usp.br/public/schMan/schMan3/macielEtAl2019/files/, was used as the reference. The filtered RNA-Seq reads were aligned with STAR v 2.7 [42] and quantified with RSEM v 1.3.1 [43], both using default parameters, and with the RSEM "estimate-rspd parameter on" option. Transcripts with counts lower than 10 were removed and differential expression analysis was performed using DESeq2 package [44] v. 1.24.0 with an FDR threshold of 0.05. PCA plot was obtained after normalization using the vst function followed by the plotPCA function from DESeq2.

## Parasite material

The BH strain (Belo Horizonte, Brazil) of *S. mansoni* was maintained in the intermediate snail host *Biomphalaria glabrata* and as the definitive host the golden Syrian hamster (*Mesocricetus auratus*). Female hamsters aged 4 weeks, freshly weaned, weighing 50–60 g, were housed in cages ($30 \times 20 \times 13$ cm) containing a sterile bed of wood shavings. Female mice (C57BL/6) aged 5 weeks, weighing 17–20 g, were housed in cages (100 cm$^2$ in a 12,7 cm height) containing a sterile bed of wood shavings. A standard diet (Nuvilab CR-1 Irradiada, Quimtia S/A, Paraná, Brazil) and water were made available *ad libitum*. The room temperature was kept at $22 \pm 2°C$, and a 12:12 h light-dark cycle was maintained.

Hamsters were infected with an *S. mansoni* cercariae suspension containing approximately 200–250 cercariae via subcutaneous injection [45]. After 42 days of infection, *S. mansoni* adult worms were recovered by perfusion of the hepatic portal system [46].

Mice (C57BL/6) were infected with an *S. mansoni* cercariae suspension containing approximately 145 cercariae via a metal ring placed on the shaved abdominal skin, for 30 min, under ketamine hydrochloride (10 mg/kg body weight) and xylazine (0.5 mg/kg body weight) (Sespo, Sao Paulo, Brazil) anesthesia. After 49 days of infection, *S. mansoni* adult worms were recovered by perfusion of the hepatic portal system [46]. *S. mansoni* eggs were extracted from *S. mansoni* infected mice livers as previously described [47].

## Parasite *in vitro* mimetic model of paired and unpaired adult worms

To study the impact of lncRNAs on the pairing status of *S. mansoni*, we have used an *in vitro* mimetic protocol in which the adult worms (*S. mansoni* males or females) retrieved from perfusion of hamsters 42 days after infection with mixed sex cercariae were cultured *in vitro* for up to 8 days either as paired couples or as separated worms [12] (**Fig 1B**). Importantly, the naturally unpaired worms retrieved from perfusion were cultured *in vitro* for mimicking the single-sex worms.

For the pairing experiments, ten adult worm paired couples, ten unpaired males or ten unpaired females naturally recovered from the hamster perfusion were used per sample and placed in each well of a 6-well plate. Among the naturally unpaired worms that were recovered in the perfusion, a fraction of approximately 5–10% were smaller female worms with less hemozoin pigment, and they were discarded; only worms that exhibited a mature phenotype upon visual inspection were used for *in vitro* unpairing assays. Three biological replicate

samples of each condition were cultured in 5 mL of ABC media [30] for 2 up to 8 days, and 70% of the media was exchanged every other day. For each time point that was assayed, a different 10-worms sample was used.

### dsRNA synthesis and *in vitro* silencing assays

To select the region of the lincRNA transcript to be targeted by the dsRNA, an *in silico* analysis of the transcript on-target and off-target bases was performed to avoid off-target effects; each lincRNA sequence was searched against the previously published *S. mansoni* protein-coding and lncRNA genes transcriptome [41] and updated by searching against the recently published version 9 protein-coding genes transcriptome (23April2022) [48] using BLASTN [49] and the following parameters: -word-size 20 -evalue 30 -num_alignments 100000 -num_descriptions 100000 -ungapped. All matching segments were discarded and only the lncRNA sequence segment with at least 200 on-target bases that did not match any other *S. mansoni* transcript and are unique to the lincRNA transcript of interest was selected for primer design and sequence amplification and cloning (**Table B in S1 Appendix**). Out of the 6 possible lincRNAs that were validated in our mimetic model by the RT-qPCR analysis, we selected 3 for *in vitro* silencing and for *in situ* hybridization (method described below): SmLINC101519, SmLINC110998, and SmLINC175062. Because SmLINC142881 has only 46 bases in its transcript that could be targeted by the dsRNA with no off-target matches, it was not chosen for silencing; while SmLINC141426 was selected only for *in situ* hybridization (see below). SmLINC133371 was discarded because it was not possible to amplify double-stranded DNA templates by PCR.

Double-stranded RNA (dsRNA) was synthesized from DNA templates amplified from cDNA of male and female adult worms, using the specific primer sequences indicated in **Table C in S1 Appendix**, all of them containing, in addition to the lincRNA sequence, a 17-nt T7 RNA Polymerase promoter sequence at their 5′-end. The *in vitro* dsRNA transcription reaction was adapted from a tRNA transcription protocol [50]. Briefly, reactions were performed at 37˚C for 12 h in a buffer containing 40 mM Tris–HCl (pH 8.0), 22 mM MgCl2, 5 mM DTT, 2 mM spermidine, 0.05% BSA, 15 mM guanosine monophosphate, 7.5 mM of each nucleoside triphosphate, amplified template DNA (0.1 µg/µL) and 5 µM of T7 RNA polymerase. The transcribed dsRNA was treated with DNase at 37˚C for 30 min and precipitated using 1:10 (v/v) 3 M sodium acetate pH 5.2 and 1:1 (v/v) of isopropanol. The pellet was washed twice with 70% ethanol and then eluted in apyrogenic saline to reach a final concentration of 3 µg/µL. Double-stranded RNAs (30 µg/mL/day) were provided to the parasites via soaking [51]. The mCherry gene was used as a non-related dsRNA control [52] and its DNA template was amplified from a pPLOT-mCherry plasmid containing the mCherry gene.

For the silencing experiments, adult worms retrieved from infected Syrian hamsters were cultured for eight days in ABC media [30] supplemented with the dsRNA targeting each of the lincRNAs of interest, with mCherry dsRNA (a control dsRNA that will activate the RNAi pathway but will not target any parasite gene [52]) or with no dsRNA. While dsRNA was added every day to the culture (to a final concentration of 30 µg/mL), the medium was exchanged every two days (70% of medium exchange). Worm pairing, adhesion, and motility were observed every day, while worm gene silencing, worm viability, worm proliferative cell status, female worm vitellaria status, and egg-related phenotypes were observed only after the eight-day treatment. At the end of the experiment on the 8[th] day, the adult worm couples were collected and stored in RNAlater (Thermo Fisher) for RNA extraction; whereas for worms labeling with thymidine analog 5-ethynyl-2′-deoxyuridine (EdU), the analog was added to the cultures on the 7th day of cultivation, incubated for 24 h and processed for microscope image acquisition as described further below.

## *In vivo* silencing

Infected mice (C57BL/6) were injected intravenously via the retro-orbital plexus with 0.1 mL of the indicated dsRNA at the concentration of 0.3 mg/mL (meaning 30 μg of dsRNA per injection), following the application of anesthetic collyrium (proxymetacaine hydrochloride Eye Drops 0.5%). The dsRNA injections occurred at weeks 1, 3, 5, and 6 post-infection.

## RNA extraction and cDNA synthesis

Total RNA from adult males and adult females was extracted using the Qiagen RNeasy Plus Micro Kit (Cat number 74034). Briefly, 20–40 adult paired and unpaired males and females for each of four biological replicates were grounded with glass beads in buffer RLT supplemented with 2-mercaptoethanol, according to Qiagen recommendation, for 2 min and then frozen in liquid nitrogen. After freezing, male and female adult worms were ground only once and frozen and thawed three times. The protocol was followed according to the manufacturer's instructions, including gDNA exclusion by the provided gDNA elimination column. All RNA samples were quantified using the Qubit RNA HS Assay Kit (Q32852, Thermo Fisher Scientific), and the integrity of RNAs was verified using the Agilent RNA 6000 Pico Kit (5067–1513, Agilent Technologies) in a 2100 Bioanalyzer Instrument (Agilent Technologies). For the pairing status experiments, complementary DNAs (cDNAs) were obtained by reverse transcription (RT) from 1000 ng of total RNA, and for the *in vitro* and *in vivo* silencing experiments, cDNA was obtained from 200 ng and 400 ng of total RNA, respectively, using SuperScript IV Reverse Transcriptase (18091050, Invitrogen) and random hexamer primers in a 20 μL volume, according to the manufacturer's recommendations.

## Primer design, quantitative RT-qPCR assays and analyses

All primer pairs were designed using the PrimerQuest Tool provided by IDT Integrated DNA Technologies (https://www.idtdna.com/PrimerQuest/) with PCR amplicon length ranging from 50 to 300 bp and melting temperature (Tm) of approximately 60˚C. All primer sequences are reported in **Table C in S1 Appendix,** and the primers efficiencies are shown in **Table D in S1 Appendix** (reference genes), **Table E in S1 Appendix** (pairing-dependent protein-coding genes), and **Table F in S1 Appendix** (pairing-dependent lncRNAs). The Cq values from the measurements of the 10 selected lincRNA expression levels on the *in vitro* mimetic model samples (at days 2, 4 and 8 of incubation) are shown in **Table G in S1 Appendix**. The Cq values obtained from the *in vitro* silencing of the 3 selected lincRNAs are described in **Table H in S1 Appendix**. The Cq values obtained from the *in vivo* silencing of the 3 selected lincRNAs are described in **Table I in S1 Appendix**.

After reverse transcription, the obtained cDNAs were diluted 8x in water. Quantitative PCR was performed using 2.5 μL of each diluted cDNA in a total volume of 10 μL containing 1× LightCycler 480 SYBR Green I Master Mix (04707516001, Roche Diagnostics), 800 nM of each primer in a LightCycler 480 System (Roche Diagnostics), and each real-time qPCR was run in three technical replicates. The PCR conditions included initial activation at 95˚C for 10 min; 45 cycles with denaturation at 95˚C for 10 s, annealing at 60˚C for 10 s, and extension at 72˚C for 20 s. A dissociation step (95˚C for 15 s, 60˚C for 1 min, 95˚C for 15 s, 60˚C for 15 s) was added at the end of the run to confirm the amplicon specificity for each gene. The quantitative RT-qPCR assays were performed following the MIQE guidelines [53–55]. The amplification efficiency for each primer was calculated using a diluted series of cDNA synthesized using 5 μg of RNA from *S. mansoni* male and female adult worms, as previously described [56].

Two different tools were used to evaluate gene expression stability of candidate reference genes using RT-qPCR data: geNorm [55] and NormFinder [53], and they were all processed as

previously described [56]. The ΔCt method [57] was used to determine the expression of the genes in all conditions tested, except for the silencing experiments. For the silencing experiments, the expression was measured in comparison to the control group (dsmCherry) and was conducted following a previously reported work [58]. The two best reference genes found in the present work (Smp_099630.1 and Smp_023150) were used to normalize the expression of the genes of interest in the *in vitro* mimetic model samples and in the *in vitro* silencing experiments. For the *in vivo* silencing results, Smp_196510.1 and Smp_101310.1 were used as the reference genes besides Smp_099630.1 and Smp_023150, based on a previous publication [56].

## Viability, pairing, adhesion, and motility measurement

The viability of *S. mansoni* adult worms after treatment with dsRNAs was determined by a cytotoxicity assay using the CellTiter-Glo Luminescent Cell Viability Assay (G7570, Promega) [59, 60]. The assay determines the amount of ATP present in freshly lysed adults; the assay signals the presence of metabolically active cells. Pairing status of the parasites was evaluated daily; only parasites with female completely outside the male's gynecophore canal were considered unpaired. Adhesion status of the parasites was evaluated daily by counting the number of females or males adhered to the plate by the ventral sucker; only parasites with no adhesion of the ventral sucker for a time longer than 10 s were considered non-adherent. Motility of the parasites was evaluated daily according to a previously determined score [61, 62]. Briefly, parasites with full body movement were scored 3, those that had partial or no movement, but were alive, were scored 1.5 and those that were dead scored 0; the treatment was considered lethal when no parasite movement was observed for longer than 2 min.

## Egg laying and egg parameters measurement

Eggs were collected from the medium in which the worm pairs (or unpaired females) were cultured. After medium collection, wells were washed with PBS to ensure complete collection of eggs. Once collected, the eggs were suspended and 40 μL aliquots were taken for counting under a bright light with 4-x magnification using a Nikon Eclipse inverted microscope. Counting was performed after eight days of *in vitro* culturing with dsRNA targeting the genes of interest and the amount of eggs/couple/day of incubation was then estimated. Images of the collected eggs were acquired under a 10-x magnification Nikon Eclipse inverted microscope, and the egg size was measured using ImageJ.

The egg shell integrity was measured based on the egg autofluorescence (which is present because of a high concentration of phenolic proteins that form the eggshell [63]). Autofluorescence was acquired by fluorescence microscopy with a 492 nm emission microscope filter under a 40x magnification using a Nikon Eclipse fluorescence inverted microscope. Egg autofluorescence was quantified using ImageJ.

Egg proliferation assays were conducted with synchronized eggs as previously reported [30]. Briefly, collected eggs were cultured *in vitro* for 7 days in ABC media. Cell proliferation was assayed by labeling with thymidine analog 5-ethynyl-2′-deoxyuridine (EdU), which was added to the eggs on the 6th day of *in vitro* culture at a final concentration of 10 μM and incubation for 24 h. EdU detection was performed as previously described [64]. Cells nuclei were stained with 10 μM of DAPI (Sigma). The ratio of proliferating cells over total cells was measured with ImageJ.

Egg hatching assays were conducted with eggs laid from females after *in vitro* culturing or with the eggs retrieved from the livers of infected mice from the *in vivo S. mansoni* lincRNAs silencing assays. Eggs were collected from livers and incubated *in vitro* for 7 days in ABC media for synchronization, as previously reported [30]. *S. mansoni* eggs extraction and hatching were conducted as previously reported [30, 47].

## Adult worm staining

Cell proliferation was assayed by labeling worms with thymidine analog 5-ethynyl-2′-deoxyur-idine (EdU), which was added to the cultures on the 7th day of cultivation at a final concentration of 10 μM and incubated for 24h. Fluorescence images were acquired on the 8th day from both male and female worms that were separated by the lincRNA silencing or that eventually remained paired. EdU detection was performed as previously described [64], with a protocol that induces the separation of worms by incubation for 1–2 min with 0.25% ethyl 3-amino-benzoate methanesulfonate (Tricaine, Sigma) anesthetic [30] at the end of the 8th day of silencing (after the 24h incubation with EdU) followed by processing for fluorescence images acquisition. EdU incorporation into the DNA of newly replicated cells was detected with click chemistry between the incorporated EdU and 10 μM Azide Fluor 545 (Sigma) under a reduced environment [30]. Detection of Azide Fluor 545 was performed with Excitation/Emission (Ex/Em) 546/565 nm. Cells nuclei were stained with 10 μM DAPI (Sigma) and detection was observed upon 405/470 nm Ex/Em. Female vitellaria was stained with Fast Blue BB and BOD-IPY C3 succinimidyl ester for vitelline and lipid droplets visualization, respectively, as previously described [30]. Detection of Fast Blue BB and BODIPY was observed upon 561/575 nm and 493/503 nm Ex/Em, respectively. DAPI was used to stain the cells nuclei, and detection was observed upon 405/470 nm Ex/Em. All images were acquired using a Confocal Microscope Leica TCS SP8, at magnification powers of 10X and 40X.

## Adult worm staining quantification

EdU positive cells from male and female adult worms and their gonads were quantified with Fiji (ImageJ) [65]. Briefly, images were transformed into 8-bit, the threshold was set to 50, and the Binary-Watershed function was applied to all images. Finally, particle analysis was processed with a delimited area of 300 μm x 300 μm for males and 250 μm x 250 μm for females, in their respective head, body and tail parts. For the adult worm gonads (testes and ovaries), the particle analysis was performed in a free-handed draw region corresponding to only their gonads. Particle analysis was set with size in inches$^2$ and circularity from 0.001 to 1.0 and excluding edges. Cell counting was manually curated and checked for each automated counting. Normalization was conducted according to the measured area in μm$^2$, and the number of EdU$^+$ cells per μm$^2$ was computed. A total of 28 images from each condition was quantified, from 4 different biological replicates (7 images per biological replicate).

Female adult worm vitellaria fluorescence staining images were acquired under the same conditions (i.e., gain) for all samples, thus quantification of Relative Fluorescence Units (RFU) can be compared. Seven delimitated areas of 50 μm x 50 μm were drawn and placed at different regions of the vitellaria to measure RFU in different segments; two squares were placed at the central region of the vitellaria corresponding to the vitelline duct and five squares were placed at distal regions of the vitellaria. RFU was measured for BODIPY (lipid droplets) and Fast Blue BB (vitelline droplets) channels in each area and the median of each measured RFU was computed. Normalization was conducted according to the measured area in μm$^2$, and RFU per μm$^2$ was plotted. A total of 28 images from each condition was quantified, from 4 different biological replicates (7 images per biological replicate).

## Whole mount in situ hybridization and imaging

To perform *in situ* hybridization experiments for lincRNA localization, 4 lncRNAs were selected based on the existence of only one transcript isoform per gene in the locus, and on the ability to design a probe that only matched a single locus in the genome. To design primers that amplify sequences unique to each lincRNA, each lincRNA sequence was searched against

the previously published *S. mansoni* protein-coding and lncRNA genes transcriptome [41] and updated by searching against the recently published version 9 of the protein-coding genes transcriptome (23April2022) [48] using BLASTN [49] and the following parameters: -word-size 20 -evalue 30 -num_alignments 100000 -num_descriptions 100000 -ungapped. All matching segments were discarded and only the lncRNA sequence segment with at least 200 on-target bases that did not match any other transcript was selected for primer design and sequence amplification and cloning. Information regarding the Gene_ID, lncRNA Transcript _ID and probe size are described in **Table B in S1 Appendix**. Pairs of primers to clone all 4 lncRNA marker probes were designed using the PrimerQuest Tool provided by IDT Integrated DNA Technologies (https://www.idtdna.com/PrimerQuest/) and are shown in **Table C in S1 Appendix**. All cloned lncRNA probe sequences were confirmed with Sanger sequencing.

The probe sequences of interest were inserted into pJC53.2 (available from Addgene https://www.addgene.org/26536/) that had been previously digested with Eam1105I. The insert orientation was confirmed with Sanger sequencing using T3 or SP6 generical primers, and the antisense *in situ* hybridization probes were synthesized accordingly, using T3 or SP6 RNA polymerase, as previously described [66, 67].

Whole mount colorimetric *in situ* hybridization analyses were performed as previously described [66, 67]. All lncRNA probes were used at 10 ng/mL in hybridization buffer. Brightfield images were acquired on a Zeiss AxioZoom V16 equipped with a transmitted light base and a Zeiss AxioCam 105 Color camera. In parallel, the negative control sense probe for each lincRNA was synthesized and tested, and the negative control WISH images are shown in **Fig F, panel B and Fig H in S1 Text**; no signal was detected with any of the sense probes tested.

### Statistical analyses

Statistically significant differentially expressed genes in the RNA-Seq analyses were determined with DESeq2 (FDR < 0.05) [44]. To determine the statistically significant differentially expressed genes in the RT-qPCR experiments, and the statistically significant changes in phenotypic parameters, all data were submitted to Normality/Log Normality tests (GraphPad Prism 9) to confirm that the data passed the test. Unpaired two-tailed Student t-test with Welch's correction was used; when indicated, One-Way Welch ANOVA test with Holm-Sidak's multiple comparisons to dsmCherry group was used (GraphPad Prism 9). The p-values are indicated in the figures' legends.

### Supporting information

**S1 Text. PDF document with Supplementary Data, Supplementary Methods, Tables N and Z, and Figs A to S.**
(PDF)

**S1 Appendix. Excel document with Tables A to M.**
(XLSX)

**S2 Appendix. Excel document with Tables O to Y.**
(XLSX)

### Acknowledgments

We would like to thank the Laboratório de Biologia Celular (Dr. Carlos Jared) from Instituto Butantan and the Confocal Lab Technician Alexsander Seixas de Souza for the services provided on the Confocal Microscope Leica TCS SP8. The WISH experiments were carried out by

Dr. Lu Zhao in the laboratory of Dr. James J. Collins III, Department of Pharmacology, UT Southwestern Medical Center, Dallas, TX 75390, USA, and we thank them for the generous gift of their work and data to be included in the present work.

## Author Contributions

**Conceptualization:** Gilbert O. Silveira, Murilo S. Amaral, Sergio Verjovski-Almeida.

**Data curation:** Gilbert O. Silveira, Lucas F. Maciel, Ana C. Tahira.

**Formal analysis:** Gilbert O. Silveira, Lucas F. Maciel, Ana C. Tahira.

**Funding acquisition:** Sergio Verjovski-Almeida.

**Investigation:** Gilbert O. Silveira, Helena S. Coelho, Adriana S. A. Pereira, Patrícia A. Miyasato, Daisy W. Santos, Giovanna G. G. Olberg, Maria Leonor S. Oliveira, Murilo S. Amaral.

**Methodology:** Gilbert O. Silveira, Murilo S. Amaral, Sergio Verjovski-Almeida.

**Project administration:** Murilo S. Amaral, Sergio Verjovski-Almeida.

**Resources:** Eliana Nakano.

**Software:** Lucas F. Maciel, Ana C. Tahira.

**Supervision:** Murilo S. Amaral, Sergio Verjovski-Almeida.

**Validation:** Gilbert O. Silveira, Murilo S. Amaral, Sergio Verjovski-Almeida.

**Visualization:** Gilbert O. Silveira, Lucas F. Maciel.

**Writing – original draft:** Gilbert O. Silveira.

**Writing – review & editing:** Murilo S. Amaral, Sergio Verjovski-Almeida.

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
