## [Decision Letter · Decision Letter 0]

22 Jan 2023

Dear Dr, Amaral,

Thank you very much for submitting your manuscript "Long non-coding RNAs are essential for Schistosoma mansoni pairing-dependent adult worm homeostasis and fertility" for consideration at PLOS Pathogens. As with all papers reviewed by the journal, your manuscript was reviewed by members of the editorial board and by three independent reviewers. In light of the reviews (below this email), we would like to invite the resubmission of a significantly-revised version that takes into account the reviewers' comments.

We cannot make any decision about publication until we have seen the revised manuscript and your response to the reviewers' comments. Your revised manuscript is also likely to be sent to reviewers for further evaluation.

Sincerely,

Christoph G. Grevelding

Guest Editor

PLOS Pathogens

P'ng Loke

Section Editor

PLOS Pathogens

Kasturi Haldar

Editor-in-Chief

PLOS Pathogens

orcid.org/0000-0001-5065-158X

Michael Malim

Editor-in-Chief

PLOS Pathogens

orcid.org/0000-0002-7699-2064

Reviewer's Responses to Questions

**Part I - Summary**

Reviewer #1: The authors tried to understand the roles of long non-coding RNAs in male and female pairing for Schistosoma mansoni. Certainly, the manuscript focuses on a potentially interestingly topic for Schistosoma. Unfortunately, the manuscript suffers from several weaknesses.

1. Line 128: As indicated,16583 lncRNAs were identified, however, it was not clear that why the authors choose the 12 lincRNAs for further analysis. Since lincRNAs usually display tissue- and temporal-specific expressions, the authors utilized the RNA-seq data published by Lu et al , in which the adult worms collected from 42-67 days of post infection were used. Although the authors tried to validate the expression of 12 selected lncRNAs, it is unclear which stage of parasites were used. Obviously, the RT-qPCR analysis of expression of lncRNAs were lack of systematic validation currently.

2. The authors used dsRNA to silence the lncRNAs and then evaluate the effect by using Edu staining, the authors claimed that cell proliferation and vitellaria composition were potentially regulated by the lncRNAs. Obviously, the Edu staining in different treatments was lack of normalization. For example, it is possible that the authors use DAPI to stain cell to normalize the Edu staining. If so, the conclusion should be stronger.

3. Line 292-297, there were several contradictory statements/observations between Wish and single cell sequence results. For examples, the authors mentioned that SmLINC101519 was expressed in the female uterus, vitellaria and around the female ovary, However, based on the Wish results, SmLINC101519 was not significantly enriched in female ovaries, the results contradicted the statements. In addition, the authors should figure out the possible reasons causing the inconsistent results between WISH and single cell sequence and then selected one confidence method to draw the related conclusions.

4. More importantly, the manuscript is lack of mechanism study of lncRNAs for regulating schistosome pairing and egg production.

5. The manuscript also suffer from poor organization, unclear descriptions and others.

Reviewer #2: The manuscript by Silveira et al. extends our knowledge on long non-coding RNAs of Schistosoma mansoni. After previous publications of the group on the expression of lncRNAs across the life-stages and across single-cells, now a re-analysis of publicly available RNAseq data of different sexes and gonads is presented. Among the lncRNAs, the focus was then set on the group of long intergenic non-coding (linc) RNAs. An impressive amount of data is presented that covers the functional characterization of several lincRNAs using basically all methods commonly used in schistosome research (RNAi, WISH, single-cell data analysis, EdU and other cell assays). The obtained results suggest a role of the selected lincRNAs in pairing stability, cell proliferation, vitellarium homeostasis and egg development. Finally, in vivo experiments using RNAi in infected mice were conducted which revealed a reduced number of worm pairs.

The study is well conducted and well written, and I am personally impressed by the amount of data. Apart from some requests for small modifications (see below), I like to raise one concern regarding the “in vitro mimetic model“ and one regarding the interpretation of pairing-dependent phenotypes:

1.) An in vitro mimetic model was used to validate the RNAseq data (Lu et al) on differential expression of selected genes between parasite sexes and pairing states in own qPCR analyses. Instead of the analysis of single-sex worms, this model involved the culture of worms, which were found separated upon perfusion, for a couple of days, in order to led them de-differentiate. This might be seen critical at first sight: it is unclear for how many days those worms were already separated in the hamster, or whether they freshly separated during perfusion. This way, worms with a different differentiation level enter the same experimental group. However, as long as the variance of gene expression between replicates is reasonable (which seems to be the case if I see Fig. 3 and 4), I would agree that this model is useful to learn about pairing-dependent processes.

2.) Seeing the high pair separation at the end of the experiment (70-90% separated, Fig. 5B), any conclusion on the effect of RNAi on the differentiation status of females (vitelline and lipid droplet staining; EdU+ cells) and their egg production is difficult. Of course, separated females will produce fewer eggs, cell proliferation will decline etc. I think one cannot be sure if the affected differentiation/egg production is just a consequence of the physical separation of females from their male partner, or if it is indeed pointing to a role of the lincRNA in the female. Can the authors defend their conclusion that a role for reproduction holds true?

If those parameters would be affected in still paired females, the interpretation would be easier. Did the authors compare the vitelline/lipid/EdU stainings of the few females that were still paired at the end of the experiment with those that were separated? Were effects similar? It should be specified in the Methods section or the figure legends of Fig. 6 and 7 whether only separated females were assessed by fluorescence microscopy, or both, paired and separated ones.

Reviewer #3: The paper identified several lncRNAs associated with pairing of Schistosoma mansoni and the authors used in culture and in vivo knock down approached to try and shed light on the possible functions of some of those lncRNAs. The data points towards involvement of these lncRNAs in several important biological processes such viability, motility, cellular proliferation in body and gonads the development of the vitellaria. Some of the lncRNAs studies could be linked with specific phenotypes, however, to the opinion of this reviewer some of the conclusions are overstated and are not directly supported by the data.

**Part II – Major Issues: Key Experiments Required for Acceptance**

Reviewer #1: (No Response)

Reviewer #2: None

Reviewer #3: Results:

Fig - 5: It appears that the knock down of both DSLinc175062 and DSLinc110998 did not reduce viability while there is a significant reduction in motility. Thus, could it be that the reduction in adhesion and pairing is simply because of reduction in motility? how can we conclude that the it is a specific effect on coupling?

It is also surprising to see that even though DSLinc124324 significantly reduced the viability of the worms, their egg production was not significantly changed? Can the authors explain this?

Along this line...is it surprising that DSLinc124324 reduced body cell proliferation (Fig 6)? This should be expected when viability is reduced. In addition, it is not clear how the data in Fig 6 was quantified? i.e. the images presented are representation of how many worms? How did the EDU+ cells were quantified and compared with the controls? Given the differences in body cell proliferation between the sexes I am wondering if the reduced viability observed for DSLinc124324 and DSLinc101519 was measured for each of the sexes? Also it appears that female body cells continue to proliferate in DSLinc101519 treatment even though the viability of this knock/down was reduced. Could it be that the reduced viability shown in Fig 5 is primarily attributed to males and that females were less affected? Sex dependent viability phenotype could contribute to the conclusions of this paper.

Fig. 7 – very nice data regarding DSLinc110998, but again since DSLinc124324 reduced viability what can we learn from the reduced vitelaria staining? Some kind of quantitative analysis is necessary.

Line 281 – “Taken together, these results show that lincRNAs are important for maintenance of adult worm pairing, viability, and fertility in vitro” this sentence is not necessarily accurate…some are important for viability other have impact on fertility and since those that do not affect viability affect motility maybe the aberrant motility is what reduces pairing?

Fig 8 – The WISH results indicated that LINC175062 is expressed in mature vitellocytes. However, Fig 6 showed that its k/d had no or minor phenotype on the vitteline droplets, this is interesting and worth discussing when speculating about the role of these lncRNAs.

**Part III – Minor Issues: Editorial and Data Presentation Modifications**

Reviewer #1: (No Response)

Reviewer #2: Line 69: The continously repeated statement that schistosomes are the only dioecious trematodes should no longer be used. Newer data identified a trematode species in crocodiles (Griphobilharzia amoena) that has a male and female individual (doi: 10.1645/GE-12-149.1). It is considered to belong to the Schistosomatoidea superfamily and to be a sister to the families of Spirorchidae and Schistosomatidae. Please rephrase this sentence (e.g. to „are the only mammalian trematodes“ or “are among the few trematodes“).

Line 25 and lines 99f: The authors emphasize that a role of lncRNAs in maintaining the pairing status and female fertility is not yet known. It don’t think this is true. In their own paper (Silveira et al. 2002, doi: 10.1007/s00436-021-07384-5) it is stated “knockdown of one lncRNA in Schistosoma mansoni, SmLINC156349, which led to … pairing impairment, … and 33% reduction in female oviposition“. Both sentences in the text should be rephrased accordingly.

Lines 142 ff (Selection of lincRNAs to be tested) Why did the authors particularly focus on the functional characterization of LINC RNAs, and not LNCA or LNCS RNAs? Was this decision based on the obtained RNAseq results? I.e. was it more likely to obtain phenotypes regarding pairing and reproduction when knocking down lincRNAs compared to the other groups? The focus on lincRNAs should be explained to the reader.

The reference list should be checked for consistency in the format of the individual references (species names should be written in italics; capitalization in article titles should be removed; etc.)

Figures

Fig. 5 and Fig. 9: Please specify in the legend for which graph which statistical test was applied (t-test or ANOVA).

Fig. 5D and E, and lines 230f: The degree of motility reduction does not match the degree of viability as measured by ATP levels. In fact knockdown of the two lncRNAs with least effects on motility led to the highest drop of ATP. How is this interpreted?

Fig. 5: It would be valuable to add some representative images of worms and eggs to the supplementary data, which depict the phenotype of the worms, the shape/size of eggs, the egg autofluorescence and EdU/DAPI staining of eggs - as a minimum from one control condition and one selected RNAi condition, to give the reader an idea how a significantly reduced parameter looks like.

Fig. 8 and other ISH figures: Were sense probes used to check for unspecific signals in ISH? Representative images of worms treated with sense probes should be added to the supplementary information or main figure. This is needed, amongst others, to correctly interpret situations where the sc expression data (no signal) do not match the ISH results (strong signals), e.g. in Fig. 8D and suppl. Fig. 3B.

Methods

The part “Parasite in vitro mimetic model of paired and unpaired adult worms“ reads like a mixed results and discussion chapter and is not fitting to a Materials & Methods chapter. Most of it should be transfered to the results and discussion part of the manuscript, or moved to the Supplementary Methods/Supplementary Data.

Line 608: (RNAi) “an in silico analysis of the transcript on-target and off-target bases was performed“ – describe how this was done, which software was used, and give a reference if possible. Was the probability of off-target knockdown of other (lnc)RNAs by the selected dsRNA sequences bioinformatically checked and excluded, e.g. by BLAST?

Line 630: Please give a reference for your selection of the mCherry gene to serve as non-related dsRNA control. Was it previously validated that mCherry dsRNA does not influence gene expression? Other typically used controls like GFP dsRNA unexpectedly have an influence, although not expressed in schistosomes.

Line 642f: “At the end of the [RNAi] experiment, the adult worm couples were collected and stored in RNAlater“. Probably some other storage conditions need to be added to the text – or were EdU stainings and vitellaria stainings done on worms that were stored in RNAlater?

Line 754: Indicate with which magnification powers the microscopic images were acquired, and which wavelengths were used for excitation of the four dyes.

Line 761: Specify which platform/tool was used (BLAST?) to search each lincRNA sequence against the previously published S. mansoni transcriptome

A chapter describing the statistical test(s) is missing in the methods section.

Supplementary Methods in S1 Text: For transparency of the lincRNA selection pipeline, it would be appreciated to add to Appendix 2 Excel tables that list the genes that are highlighted with circles in Figures B-E and those indicated in Table B.

Reviewer #3: Line 98- the difference between lncRNA and lincRNA should be defined

PLOS authors have the option to publish the peer review history of their article (what does this mean?). If published, this will include your full peer review and any attached files.

Reviewer #1: No

Reviewer #2: No

Reviewer #3: No
---

## [Decision Letter · Decision Letter 1]

18 Apr 2023

Dear Dr. Amaral,

We are pleased to inform you that your manuscript 'Long non-coding RNAs are essential for Schistosoma mansoni pairing-dependent adult worm homeostasis and fertility' has been provisionally accepted for publication in PLOS Pathogens.

Best regards,

Christoph G. Grevelding

Guest Editor

PLOS Pathogens

P'ng Loke

Section Editor

PLOS Pathogens

Kasturi Haldar

Editor-in-Chief

PLOS Pathogens

orcid.org/0000-0001-5065-158X

Michael Malim

Editor-in-Chief

PLOS Pathogens

orcid.org/0000-0002-7699-2064

The authors addressed the points of all reviewers in a satisfying manner. Therefore, I recommend to accept the paper for publication.

Reviewer Comments (if any, and for reference):

Reviewer's Responses to Questions

**Part I - Summary**

Reviewer #1: The authors tried to address the reviewer's comments in the revised manuscript, so I suggest it to be accepted for publications.

Reviewer #2: The authors addressed the previous concerns of the reviewers in a very comprehensive way. I have no further additions to that, well done.

Reviewer #3: The authors have thoroughly revised the paper following my comments. This is a nice piece of work and i have no further comments.

**Part II – Major Issues: Key Experiments Required for Acceptance**

Reviewer #1: (No Response)

Reviewer #2: (No Response)

Reviewer #3: (No Response)

**Part III – Minor Issues: Editorial and Data Presentation Modifications**

Reviewer #1: (No Response)

Reviewer #2: (No Response)

Reviewer #3: (No Response)

PLOS authors have the option to publish the peer review history of their article (what does this mean?). If published, this will include your full peer review and any attached files.

Reviewer #1: No

Reviewer #2: No

Reviewer #3: No

---

## [Editor Report · Acceptance letter]

28 Apr 2023

Dear Dr. Amaral,

We are delighted to inform you that your manuscript, "Long non-coding RNAs are essential for Schistosoma mansoni pairing-dependent adult worm homeostasis and fertility," has been formally accepted for publication in PLOS Pathogens.

Best regards,

Kasturi Haldar

Editor-in-Chief

PLOS Pathogens

orcid.org/0000-0001-5065-158X

Michael Malim

Editor-in-Chief

PLOS Pathogens

orcid.org/0000-0002-7699-2064